# A Novel Drug with Potential to Treat Hyperbilirubinemia and Prevent Liver Damage Induced by Hyperbilirubinemia: Carbon Dots Derived from *Platycodon grandiflorum*

**DOI:** 10.3390/molecules28062720

**Published:** 2023-03-17

**Authors:** Rui Chen, Huagen Ma, Xiaopeng Li, Meijun Wang, Yunbo Yang, Tong Wu, Yue Zhang, Hui Kong, Huihua Qu, Yan Zhao

**Affiliations:** 1School of Traditional Chinese Medicine, Beijing University of Chinese Medicine, Beijing 100029, China; 2School of Chinese Materia Medica, Beijing University of Chinese Medicine, Beijing 100029, China; 3School of Life Sciences, Beijing University of Chinese Medicine, Beijing 100029, China; 4Center of Scientific Experiment, Beijing University of Chinese Medicine, Beijing 100029, China

**Keywords:** carbon dots, *Platycodon grandiflorum*, hyperbilirubinemia, liver protection, drug therapy

## Abstract

*Platycodon grandiflorum* (PG) is a traditional Chinese medicine with a long history, but its active compounds have not been reported. In this study, novel carbon dots (CDs), PG-based CDs (PGC-CDs), were discovered and prepared from PG via calcinations and characterized by transmission electron microscopy; high-resolution transmission electron microscopy; X-ray diffraction, fluorescence, ultraviolet-visible, and Fourier-transform infrared spectrometers; X-ray photoelectron spectroscopy; and high-performance liquid chromatography. In addition, the safety and antioxidant activity of PGC-CDs was evaluated by RAW264.7 cells and LO2 cells. The therapeutic effects of PGC-CDs on hyperbilirubinemia and liver protection were evaluated in a bilirubin-induced hyperbilirubinemia mice model. The experiment confirmed that the diameter range of PGC-CDs was from 1.2 to 3.6 nm. PGC-CDs had no toxicity to RAW264.7 cells and LO2 cells at a concentration of 3.91 to 1000 µg/mL and could reduce the oxidative damage of cells caused by H_2_O_2_. PGC-CDs could inhibit the increase levels of bilirubin and inflammation factors and increase the levels of antioxidants and survival rate, demonstrating that PGC-CDs possessed anti-inflammatory and anti-oxidation activity. PGC-CDs may reduce the content of bilirubin, so as to reduce a series of pathological lesions caused by bilirubin, which has potential in treating hyperbilirubinemia and preventing liver damage induced by hyperbilirubinemia.

## 1. Introduction

As a Chinese herbal medicine with the homology of medicine and food, the medicinal *Platycodon grandiflorum* (PG) taken from the root of the plant PG was first recorded in Shennong Herbal Scripture and widely used in China. As is known to all, the way Chinese herbal medicine is processed affects its efficacy in most cases, and its chemical composition and pharmacological action may change with the change of processing method. Likewise, the medicinal constituents of PG can be affected by the processing temperature and time [1]. As a matter of fact, as early as more than 1000 years ago, the earliest record of charring PG in China was found in *Zhou Hou Bei Ji Fang*, which was the first book recording the processing method of burning crushed PG and treating coma with PG Carbonisata (PGC). Later, many studies have found that the medical use of PGC was expanded, and it can be used to treat inflammation, metabolic diseases and so on. However, the material basis and mechanism of PG after carbonization are still unclear.

In recent years, carbon dots (CDs) with unique advantages, such as water solubility, good biocompatibility and photochemical stability, and low toxicity or non-toxicity, have attracted great attention in many fields, especially in the biomedical field. Due to the many values of CDs in the biomedical field, the biological activities of CDs have been gradually explored and reported, such as antibacterial [2], antiviral [3], anti-tumor [4], anti-oxidation [5], anti-allergy [6] and immunomodulatory [7] activities. These good biological activities suggest the broad prospects of CDs in the field of biopharmaceutics. As a natural medicine, Chinese herb has a wide range of development and research value. At present, many researchers have extracted CDs from Chinese herbal medicine by calcinations and demonstrated the fascinating bioactivity, such as the hypoglycemic effect of Jiaosanxian-derived CDs [8], the kidney protection effect of Phellodendri Chinensis Cortex Carbonisata CDs [9], the brain protection effect of Semen pruni persicae and Carthamus tinctorius L.-derived CDs [10], the thermoregulation effect of Lonicerae japonicae Flos Carbonisatas CDs [11], the anti-inflammatory effect of Mulberry silkworm cocoon CDs [12], the anti-frost effect of Artemisiae Argyi Folium Carbonisata CDs [13], the anti-anxiety and calming effect of cigarette mainstream smoke CDs [14] and so on. Although CDs have been extracted from many Chinese herbs so far and proved to have considerable biological activity, there is still no research to prove whether the CDs derived from PG (PGC-CDs) have biological activity, so it is worth further exploration and research.

Hyperbilirubinemia is characterized by bilirubin beyond the normal range, which is harmful to the human body and has high morbidity and mortality [15]. There are many etiological factors, such as genetic factors, liver factors, gastrointestinal factors and so on [16]; the specific pathogenesis is not clear, which also increases the difficulty of its treatment. Remarkably, a high concentration of free bilirubin can easily pass through the blood–brain barrier and enter the brain, causing excitotoxicity, promoting the release of free radicals and pro-inflammatory factors, leading to plasma membrane disturbance, oxidative stress, neuritis and other pathological changes, making nerve cells denature and causing apoptosis, resulting in nerve dysfunction eventually [17,18,19]. It may also leave neurological sequelae that can seriously affect quality of life and, more critically, cause death from nuclear jaundice [20,21,22,23]. Currently, the treatment methods for hyperbilirubinemia mainly include anti-inflammatory or bilirubin-lowering drugs, phototherapy and blood exchange. However, most patients do not respond well to these therapies or experience adverse reactions. Therefore, the improvement of phototherapy and blood exchange therapy, as well as the search for new effective drugs, have become the focus of current research. Studies have shown that excessive bilirubin deposited in the body can also cause liver dysfunction [24]. The liver is an important organ for transforming and excreting bilirubin. Studies have shown that patients with severe bilirubin elevations may need a liver transplant [25]. In order to minimize the harm of hyperbilirubinemia to the human body, it is necessary to establish an overall plan to protect the potentially damaged liver and other important organs while preventing and treating nerve damage, such as bilirubinemia, in clinical practice. In recent years, with the development of nanotechnology, fluorescence probes [26] and kits [27] made by CDs were used for bilirubin detection, realizing the first application of CDs in the field of hyperbilirubinemia. However, it has been observed that no drugs have been developed to treat hyperbilirubinemia by taking advantage of CDs, and this study tries to break through the gap in this field.

In this study, we first obtained PGC-CDs from carbonized PG and characterized it, naming it as PGC-CDs. In addition, the safety and antioxidation activity of PGC-CDs was evaluated in cells. It was found that the serum bilirubin level could be decreased by increasing the MRP2 content in the liver of rats [28]. Therefore, by using the bilirubin-induced hyperbilirubinemia mice model, the therapeutic effect of PGC-CDs on hyperbilirubinemia and the preventive effect on liver damage induced by hyperbilirubinemia was evaluated, and the mechanism was discussed.

## 2. Results

### 2.1. Characterization of PGC-CDs

Transmission electron microscope (TEM) images (Figure 1A) exhibited PGC-CDs had monodispersed and uniform distribution, with particle sizes ranging from 1.2 nm to 3.6 nm (Figure 1A); the average diameter was 2.3 nm, and the lattice spacing was 0.228 nm (Figure 1D). X-ray Diffraction (XRD) patterns (Figure 1B) showed that there was an obvious diffraction peak at 2θ = 22,765°, which was attributed to the fact that the PGC-CDs consisted of randomly arranged amorphous carbons [29].

As can be seen from the fluorescence spectrum in Figure 1C, the maximum excitation wavelength and emission wavelength of PGC-CDs were 321 nm and 432 nm, respectively. The ultraviolet-visible (UV-Vis) spectrum of PGC-CDs (Figure 1E) showed that there was a slight absorption peak at 218 nm that may be caused by the π–π* transition of the conjugated C=C bond.

As can be seen from Figure 1F, the characteristic peaks of PGC-CDs in the Fourier transform infrared (FTIR) spectrum were distributed at 3440, 2920, 2851, 1629, 1384, 1057 and 556 cm^−1^. The characteristic peak at 3440 cm^−1^ suggested the possibility of the stretching vibration of the –O–H bond or –N–H bond, while the absorption peaks at 2920 and 2851 cm^−1^ were caused by the stretching vibration of the C–H bond of CH_2_ on PGC-CDs’ surface, and the characteristic peak at 1629 cm^−1^ was generally considered to be caused by a C=O bond [30]. The stretching vibration peak of C–N was usually located at the characteristic peak of 1384 cm^−1^, and the characteristic peak of 1057 cm^−1^ was attributed to the stretching vibration of the C–O–C bond [31,32,33].

The element composition and surface-active group of PGC-CDs were obtained by observing X-ray photoelectron spectroscopy (XPS). As can be seen from Figure 2A, there were 3 obvious peaks at 284.76, 531.83 and 399.36 eV, confirming the presence of C (75.04%), O (22.12%) and N (2.84%). The spectrum of C1 (Figure 2B) was divided into 3 peaks at 284.77, 286.32 and 288.34 eV, corresponding to C–OH, C–O and C=O. The O1s spectrum (Figure 2C) had 3 peaks at 531.66, 533.02 and 533.95 eV, which belong to C–O, C=O and H–O–H, respectively. Furthermore, Figure 2D showed 2 peaks of the N1s spectrum at 399.79 and 400.39 eV, which were attributed to C–N–C and N–H, respectively. The data of XPS determined that rich functional groups, such as carboxyl, hydroxyl and amine groups, existed on the surface of the PGC-CDs, which was similar to the results of FTIR.

The high-performance liquid chromatogram (HPLC) results of the PGC and PGC-CDs aqueous solutions are shown in Figure 3. PGC contained abundant small molecules (Figure 3A). No obvious characteristic peak was found in the HPLC chromatogram of PGC-CDs (Figure 3B), indicating that no active small-molecule compounds were identified in the PGC-CDs, so the influence of small-molecule compounds on subsequent experimental results can be excluded.

### 2.2. Cellular Toxicity

The safety of drugs is a prerequisite and necessary condition for their clinical application. In this study, the safety of PGC-CDs at different concentrations (1000~3.91 μg/mL) was evaluated by CCK-8 assay. As shown in Figure 4A, when PGC-CDs were not given, the cell viability was calculated to be 100% according to the cell viability formula, which was the normal viability of RAW264.7 cells. As a reference, when the value of cell viability was less than 100%, this indicated that it could inhibit cell proliferation; when the value of cell viability was equal to or more than 100%, this indicated that PGC-CDs did not inhibit cell growth and even promoted cell proliferation. The results showed that PGC-CDs at different concentrations did not inhibit the cell viability of RAW264.7 cells. On the contrary, PGC-CDs showed the ability to improve cell viability at a concentration from 1000 to 3.91 μg/mL, which proved that PGC-CDs had no cytotoxicity at the concentration of <1000 μg/mL. PGC-CDs did not affect the cell viability of LO2 cells at a concentration from 1000 to 3.91 μg/mL (Figure 4B). The safety of PGC-CDs provided a basis for subsequent studies on its biological activity.

### 2.3. Effect of PGC-CDs on H_2_O_2_-Induced RAW264.7 Cells

H_2_O_2_ can directly enter the cell membrane and cause oxidative damage [34]. Different concentrations of H_2_O_2_ have certain inhibitory effects on the growth of RAW264.7 cells, and the cell viability decreased with the increase of H_2_O_2_ concentration (Figure 4C). Compared with the control group, different concentrations of H_2_O_2_ significantly inhibited the survival of RAW264.7 cells (*p* < 0.05). When treated with 0.4 mmol/L H_2_O_2_, the survival rate of RAW264.7 cells reached 50.36%. Therefore, 0.4 mmol/L H_2_O_2_ was used to establish the cellular oxidative stress model. According to Figure 4A, PGC-CDs at 250, 125 and 62.5 μg/mL were selected to study its effect on H_2_O_2_-induced RAW264.7 cells. SOD and MDA are commonly used to evaluate the antioxidant effect. As shown in Figure 4D, compared with the model group (26.58 ± 3.45 U/mgprot), 250, 125 and 62.5 μg/mL of PGC-CDs could significantly increase the SOD activity (45.15 ± 5.47, 60.78 ± 6.61 and 57.14 ± 9.11 U/mgprot, *p* < 0.01). As shown in Figure 4E, compared with the control group (2 ± 0.45 nmol/mgprot), the content of MDA in the cells of the model group (17.78 ± 2.12 nmol/mgprot) increased significantly (*p* < 0.01). A total of 250, 125 and 62.5 μg/mL of PGC-CDs (12.68 ± 1.63, 8.98 ± 2.60 and 11.16 ± 3.15 nmol/mgprot, *p* < 0.01) significantly inhibited the increase of the content of MDA in cells caused by H_2_O_2_. These results indicated that PGC-CDs reduced the oxidative damage of cells and showed an antioxidant effect in cells.

### 2.4. Effects of PGC-CDs on General Condition

Before modeling, mice in each group were in a good mental state, with stable breathing and normal activities. All mice were sensitive to external stimulation and did not have abnormal behaviors, such as tremor and prone position. After intraperitoneal injection of bilirubin for 4 h, the mice in the bilirubin group (Bil group) showed poor spirit, almost no activity and obvious tremor in limbs. Eight hours after modeling, the mice had slow breathing, dark yellow urine, decreased response to external stimuli and body fibrillation. Twelve hours after modeling, the mice lay prostrate with lusterless hair. Twenty-four hours after modeling, part of their skin began to yellow, and the stool volume of the mice decreased. Forty-eight hours after modeling, the hair of the mice was visibly messy and dull. Compared with the Bil group, the general condition of the mice in the PGC-CDs group was significantly improved. The body weight (Figure 5A) and food intake (Figure 5B) of the mice within 1 week after modeling were observed and counted, and the survival rate (Figure 5C) was calculated. Compared with the Bil group, the body weight, food intake and survival rate of the mice in the PGC-CDs groups were significantly increased, and the differences were statistically significant.

### 2.5. Effects of PGC-CDs on Neurological Function

The neurobehavioral function score is often positively correlated with nerve injury and can assist in evaluating the efficacy of therapeutic drugs. Tarlov scores have been widely used after being improved by scholars [35]. In many studies [36,37], they were used to evaluate the neurological function of animals by observing the motor status of the limbs. The Tarlov scores of mice in the control group, Bil group and PGC-CDs groups from 4 h to 48 h after modeling are shown in Table 1. Before the establishment of the model, the activities of mice in each group were normal, and the Tarlov scores were 4. After intraperitoneal injection of bilirubin, the Tarlov score of the Bil group was significantly lower than that of the control group, and it was the lowest score at 24 h after modeling, indicating that the nerves of hyperbilirubinemia mice were damaged to a certain extent. It is worth noting that the Tarlov scores of the PGC-CDs groups were significantly higher than that of the Bil group, indicating that PGC-CDs can effectively improve hyperbilirubinemia.

### 2.6. Effects of PGC-CDs on Biochemical Levels

The protective effect of PGC-CDs on liver damage induced by hyperbilirubinemia was evaluated by detecting levels of DBIL, IBIL, TBIL, TBA, ALT and AST. At 24 h after the establishment of the hyperbilirubinemia model, the levels of DBIL (Figure 6A), IBIL (Figure 6B) and TBIL (Figure 6C) in the Bil group (2.13 ± 0.49, 6.61 ± 1.07 and 8.75 ± 1.52 μmol/L, *p* < 0.01) were significantly higher than those in the control group (0.37 ± 0.11, 1.15 ± 0.30 and 1.52 ± 0.27 μmol/L). It indicated that the model of hyperbilirubinemia was established successfully. Compared with the Bil group, the level of DBIL in the high-, medium- and low-dose PGC-CDs groups were 2.12 ± 0.39, 1.71 ± 0.49 and 1.73 ± 0.17 μmol/L, respectively. Compared with the Bil group, the level of IBIL in the low-dose PGC-CDs (5.16 ± 0.45 μmol/L, *p* < 0.05) was significantly decreased; however, there was no significant reduction in IBIL in the high-dose and medium-dose PGC-CDs (6.25 ± 0.94 and 5.73 ± 1.42 μmol/L). The TBIL of the low-dose PGC-CDs group (6.89 ± 0.58 μmol/L, *p* < 0.05) was significantly lower than that of the Bil group, and the high- and medium-dose PGC-CDs groups (8.38 ± 1.27 and 7.44 ± 1.88 μmol/L) did not significantly reduce the levels of TBIL. Figure 6D shows the effect of PGC-CDs on the TBA of liver damage induced by hyperbilirubinemia, which was significantly higher in the Bil group (13.50 ± 2.22 μmol/L, *p* < 0.01) than the control group (1.80 ± 0.45 μmol/L). In sharp contrast, the high-, medium- and low-dose PGC-CDs groups (8.23 ± 2.91, 6.12 ± 1.93 and 7.41 ± 1.71 μmol/L, *p* < 0.01) significantly reduced levels of TBA, and PGC-CDs significantly alleviated TBA elevation. Levels of ALT and AST can reflect the extent of liver damage. ALT (Figure 6E) and AST (Figure 6F) in the Bil group (579.83 ± 164.10 and 827.65 ± 178.66 U/L, *p* < 0.01) were significantly higher than those in the control group (29.86 ± 4.45 and 105.77 ± 10.57 U/L), indicating that the model caused serious liver damage. It is worth noting that the elevated ALT was significantly inhibited in the high and medium doses of the PGC-CDs groups (245.17 ± 98.34 and 218.50 ± 57.39 U/L, *p* < 0.05), but the low-dose PGC-CDs group (467.33 ± 289.02 U/L) showed no significant decrease. The increase of AST was significantly inhibited in the high- and medium-dose PGC-CDs groups (444.17 ± 145.06 and 488.27 ± 122.28 U/L, *p* < 0.05), and the level of AST was also decreased in the low-dose group (658.43 ± 342.65 U/L), but the difference was not statistically significant. The results indicated that PGC-CDs had the ability to prevent liver damage induced by hyperbilirubinemia.

### 2.7. Effects of PGC-CDs on Inflammatory Factors

Levels of IL-6 and TNF-α reflect the inflammatory state of the body. To investigate the effect of PGC-CDs on inflammatory factors of liver damage induced by hyperbilirubinemia, we tested the content of IL-6 and TNF-α. As shown in Figure 7A, B, the levels of IL-6 and TNF-α in the Bil group (23.09 ± 2.34 pg/mL, *p* < 0.01, 121.14 ± 5.08 ng/L, *p* < 0.05) were visibly higher than those in the control group (13.39 ± 4.17 pg/mL, 79.62 ± 18.61 ng/L), indicating that liver damage induced by hyperbilirubinemia may stimulate the production of inflammatory factors. Figure 7A shows that the high-, medium- and low-dose PGC-CDs groups (17.79 ± 2.10, 17.43 ± 1.52 and 16.20 ± 1.20 pg/mL, *p* < 0.01) could significantly inhibit the increase of IL-6. As shown in Figure 7B, the medium-dose PGC-CDs reduced the sharply elevated level of TNF-α (98.68 ± 4.50 ng/L, *p* < 0.01), while both the high- and low-dose groups (111.73 ± 3.60 and 108.46 ± 5.40 ng/L, *p* < 0.05) could reduce the increase of TNF-α. The results suggested that PGC-CDs can improve inflammation by lowering levels of inflammatory factors.

### 2.8. Effects of PGC-CDs on SOD, MDA, GSH and CAT Levels in Liver Tissue

SOD can remove free radicals and reduce the burden on the liver. As shown in Figure 7C, SOD activity in the Bil group (225.78 ± 83.31 U/mgprot, *p* < 0.01) was significantly lower than that in the control group (486.86 ± 124.07 U/mgprot). Compared with the Bil group, SOD activity in the medium- and low-dose PGC-CDs groups (472.82 ± 70.57 and 407.25 ± 53.35 U/mgprot, *p* < 0.01) was significantly increased, while SOD activity in the high-dose group (302.71 ± 100.76 U/mgprot) was also increased. The level of MDA reflects the function of the liver. As shown in Figure 7D, compared with the control group (2.19 ± 0.74 nmol/mgprot), the level of MDA in the Bil group (5.04 ± 0.94 nmol/mgprot, *p* < 0.01) was significantly increased, indicating that the liver function was seriously damaged. Levels of MDA in all doses of the PGC-CDs groups (3.04 ± 1.08, 2.56 ± 1.47 and 2.61 ± 0.76 nmol/mgprot, *p* < 0.01) were lower than that in the Bil group. GSH is directly or indirectly involved in many life activities of the body and has a protective effect on the body. As shown in Figure 7E, changes in the level of GSH are shown. Compared with the control group (102.71 ± 31.18 μmol/gprot), the level of GSH in the Bil group (25.80 ± 10.09 μmol/gprot, *p* < 0.01) was significantly decreased, while the increase of GSH in the high-dose PGC-CDs group (43.70 ± 26.98 μmol/gprot) was not significant, showing no statistical significance. Levels of GSH in the medium- and low-dose PGC-CDs groups (77.64 ± 24.18 and 74.65 ± 9.07 μmol/gprot, *p* < 0.01) notably increased the level of GSH. CAT can eliminate harmful hydrogen peroxide and protect the liver. As shown in Figure 7F, the level of CAT in the Bil group (6.38 ± 2.17 U/mgprot, *p* < 0.01) was significantly reduced compared with the control group (61.04 ± 12.26 U/mgprot). Notably, levels of CAT were significantly increased in all doses of the PGC-CDs groups (21.46 ± 6.78, 58.48 ± 8.80 and 51.57 ± 9.65 U/mgprot, *p* < 0.01). These results suggested that PGC-CDs had a good effect on antioxidants and preventing liver damage induced by hyperbilirubinemia.

These studies have shown that PGC-CDs played an extremely important role in improving general conditions, reducing the level of bilirubin, enhancing liver function, resisting inflammation and oxidative damage, and improving the survival rate.

### 2.9. Histopathological Analysis

Hematoxylin and eosin (H&E) staining was used to observe the effect of PGC-CDs on liver tissue (Figure 8). No abnormal pathological changes were found in the liver tissue of the control group. In the Bil group, under the low-power (Figure 8A) and medium-power microscope (Figure 8B), there was severely damaged liver tissue and a disordered hepatic cord, with a large amount of cholestasis and infiltration of inflammatory cells. Significantly enlarged and binucleated hepatocytes, or even necrotic hepatocytes, were seen with the high-power microscope (Figure 8C). In contrast, the pathological damage was significantly improved in the PGC-CDs groups, cholestasis and inflammatory cell infiltration were reduced apparently, and the shape and size of liver cells were significantly improved.

## 3. Discussion

In traditional Chinese medicine, PGC is a kind of traditional medicine prepared by calcinations processing. Compared with PG, PGC contains fewer volatile components, and its pharmacological activity has changed accordingly. However, the material basis of its efficacy is not clear at present. Some scholars have found that the compounds contained in herbal medicines in the process of high-temperature processing can be transformed into CDs through dehydration, calcinations and carbonization, which have different biological activities from the original medicinal materials [38,39,40]. Therefore, in this study, we successfully obtained PGC-CDs from PGC and discovered PGC-CDs were approximately spherical, with an average particle size of 2.3 nm, and mainly contained C, O and N elements, as well as carboxyl, hydroxyl, amino and other functional groups, indicating that PGC-CDs have better water solubility, more uniform particle size and new pharmacological activity different from the original medicinal compound. In addition, its safety was verified by cytotoxicity assay. Meanwhile, the antioxidant activity of PGC-CDs in cells was confirmed. The main elements of plant part-derived CDs are C, O, H and N atoms, which present in various functional groups and provide good water solubility [41,42]. In particular, doping of elements such as N would give different activity to CDs [12,41]. CDs prepared by different methods, reaction conditions and raw materials often have different properties; CrCi-CDs performs a neuroprotective effect on cerebral ischemia and reperfusion injury [41]. CDs synthesized from soybean milk not only have good photoluminescence properties, but also good electrocatalytic activity for oxygen reduction reactions [43]. Differences between PGC-CDs and other CDs were observed in terms of their structural features, optical characteristics and different biological activities [10,38,40,44], which may be due to the fact that their precursors consist of different compounds [45], and the carbon backbone condensed upon heating differs from the surface moieties.

The etiology of hyperbilirubinemia Is very complex; It is one of the most common diseases of newborns, and it also occurs in people in various conditions, such as hepatitis, cirrhosis, sepsis, liver transplantation and heart surgery [46]. Studies [46,47,48,49,50,51,52] have shown that hyperbilirubinemia and liver injury interact with each other. When liver clearance is low [53], hyperbilirubinemia may occur. Meanwhile, patients with hyperbilirubinemia may need liver transplantation [54,55]. Therefore, liver protection is particularly important in the treatment of hyperbilirubinemia. Clinically, the late manifestations of neonatal hyperbilirubinemia are developmental delay, cognitive impairment and behavioral and psychiatric disorders [56]. Studies have reported that neonates with inadequate energy intake or significant weight loss were at high risk for severe hyperbilirubinemia, while adequate feeding and a low percentage of weight loss significantly reduced the risk of jaundice [57,58]. In addition, comprehensive nursing could promote the physical and mental development of neonates with hyperbilirubinemia [59]. Therefore, continuous monitoring of body weight, food intake and neurological function and effective interventions are important for the prevention and treatment of hyperbilirubinemia. It was found that PGC-CDs had a good inhibitory effect on hyperbilirubinemia by studying the body weight, food intake, survival rate and neurological function of mice. This might be related to the electrostatic adsorption of CDs [60] and the anti-inflammatory and antioxidant effects of PGC-CDs, which could reduce the damage of excessive bilirubin on mice and improve their general state.

The treatment of hyperbilirubinemia should solve the state of high levels of bilirubin as soon as possible and reduce the content of bilirubin, which is toxic to humans. In this study, the serum biochemical results of DBIL, IBIL and TBIL increased sharply, indicating the successful establishment of a hyperbilirubinemia model. PGC-CDs can significantly reduce the level of serum bilirubin and TBA, which may be related to the electrostatic adsorption of CDs [60]. In addition, as a 1.2 nm to 3.6 nm macromolecular component with abundant surface groups, PGC-CDs may have a similar effect to macromolecular albumin and can bind to bilirubin, thus reducing the pathological damage of bilirubin to the liver. Moreover, the therapeutic effects of medium and low doses of PGC-CDs were better than those of high doses of PGC-CDs, which may be due to the fact that the particle density of PGC-CDs in a high dose was large, the aggregation and sedimentation were increased, the dispersion and fluidity were poor, and the activity was reduced [61]. Patients with hyperbilirubinemia often have higher levels of ALT and AST, and it is always necessary to require hepatoprotective therapy. ALT and AST increased sharply in the Bil group, indicating that hyperbilirubinemia can cause liver damage. It was a remarkable fact that all doses of PGC-CDs reduced the levels of ALT and AST, suggesting that PGC-CDs had a protective effect on liver damage induced by hyperbilirubinemia. This result was particularly important because the protective effect of PGC-CDs on the liver was achieved without affecting the levels of total bilirubin in plasma. Of course, pathological observations further confirmed this result.

Some chronic inflammation can cause hyperbilirubinemia [62,63]. The use of anti-inflammatory drugs can effectively reduce neonatal hyperbilirubinemia mortality and avoid nerve damage [20]. Levels of inflammatory factors in the Bil group increased sharply, suggesting that mice were in a state of severe inflammation, which inevitably affected liver function and bilirubin metabolism. It was noteworthy that all doses of PGC-CDs could significantly reduce levels of IL-6 and TNF-α, showing great anti-inflammatory activity. Inflammation caused by elevated bilirubin increases mortality from the disease. The results of this study suggested that PGC-CDs could inhibit levels of IL-6 and TNF-α in the treatment of hyperbilirubinemia and its associated liver damage, thereby reducing the mortality of hyperbilirubinemia. From the mechanism analysis, PGC-CDs could inhibit inflammation, increase hepatic blood flow, reduce edema, inhibit vascular activity and stabilize the cell membrane and lysosomal membrane. It could also reduce tissue edema, reduce the damage of tissue cell structure and inhibit the release of inflammatory factors, so as to enhance the body’s tolerance of hyperbilirubinemia.

In fact, oxidative stress reactions are common in the presence of excessive bilirubin [64,65]. Levels of SOD, GSH, CAT and other substances involved in scavenging reactive oxygen free radicals in the body decrease, leading to the increase of the content of active oxygen free radicals in the body, which in turn cause oxidative stress and damage. The level of MDA usually increases when liver cells or tissues are damaged. The results of this study showed that the activity of SOD, the level of GSH and CAT of mice in the Bil group, which decreased significantly, while the level of MDA increased, suggesting that the liver suffered more serious oxidative stress after hyperbilirubinemia. After PGC-CDs pre-treatment and SOD activity, the level of GSH and CAT increased obviously, while the level of MDA decreased significantly, indicating that PGC-CDs could effectively improve the antioxidant capacity of the body. From the mechanism analysis, PGC-CDs could antagonize the excitotoxic response of glutamate to hepatocytes, could restore the reduced antioxidant enzymes in red blood cells to the normal level and played a protective role. At the same time, they could reduce the generation of reactive oxygen species and reduce the function of mitochondrial overload, so as to exert antioxidant effect.

The hyperbilirubinemia model Induced by the direct Injection of bilirubin Is a stable, economical and simple model [49]. This study provided a reference for the establishment of a liver damage model induced by hyperbilirubinemia. More importantly, this study revealed that lowering bilirubin levels and controlling inflammation and oxidative stress may be a potential strategy to reduce the harm of hyperbilirubinemia. However, this study is only a preliminary exploration of the effects and mechanisms of PGC-CDs in the treatment of hyperbilirubinemia and its induced liver damage, and further studies are needed to clarify the deeper potential mechanisms of these effects.

## 4. Materials and Methods

### 4.1. Materials

PG was purchased from Beijing Qiancao Herbal Pieces Co., Ltd. (Beijing, China). PGC-CDs were prepared in our laboratory, and bilirubin was purchased from Yuanye Biotechnology Co., Ltd. (Shanghai, China). Other chemical reagents were purchased from Sinopharm Chemical Reagent Co., Ltd. (Beijing, China). Cell Counting Kit-8 (CCK-8) was purchased from Dojindo Molecular Technology Co., Ltd. (Kumamoto, Japan). Mice interleukin-6 (IL-6) and tumor necrosis factor-α (TNF-α) enzyme-linked immunosorbent assay (ELISA) kits were purchased from Kete Biology Co., Ltd. (Yancheng, Jiangsu, China). Protein concentration kit was purchased from Solebo Technology Co., Ltd. (Beijing, China). Malondialdehyde (MDA), superoxide dismutase (SOD), glutathione (GSH) and catalase (CAT) detection kits were purchased from Jiangcheng Institute of Biological Engineering (Nanjing, China). Deionized water (DW) was used throughout the experiment.

### 4.2. Animals

The experimental design and protocol were carried out in accordance with the approval of the Animal Experiment Ethics Committee of Beijing University of Chinese Medicine, and the management of experimental animals followed the Regulations of the People’s Republic of China on the Management of Laboratory Animals and other relevant rules and regulations. Kunming male mice (body weight 30.0 ± 2.0 g) were purchased from the Laboratory Animal Center, Si Peifu (Beijing, China). The feeding environment was suitable, and the animals were kept at a temperature of 24.0 ± 1.0 °C, a relative humidity of 55–65% and a 12 h light/12 h dark cycle, and allowed libitum feed and water. Twelve hours before the hyperbilirubinemia model was established, all animals were fed no food except water.

### 4.3. Preparation of PGC-CDs

The preparation process of PGC-CDs can be divided into carbonization, boiling and purification (Figure 9). In total, 400 g of PG was weighed and placed in a clean and dry crucible, then it was sealed with aluminum foil and put in a muffle furnace (TL0612, Beijing Zhong Ke Aobo Technology Co., Ltd., Beijing, China) for calcinations. The temperature process of the muffle furnace was as follows: the calcined temperature was increased to 70 °C within 5 min; after maintaining at 70 °C for 30 min, the temperature was increased from 70 °C to 350 °C within 25 min, and maintained at 350 °C for 1 h. After the temperature of the muffle furnace dropped to 40 °C, the PGC was further taken and crushed. Then the PGC power was added to thirtyfold DW and boiled at 100 °C twice for 1 h, during which a glass rod was used to stir evenly. The decoction was combined and filtered with an 0.22 μm microporous membrane, and the filtrate was collected and concentrated by a rotary evaporator to obtain 1 g/mL PGC solution. The concentrated solution was transferred into a dialysis membrane with a molecular weight cut-off of 1000 Da, and the dialysis membrane was placed in a beaker with DW for 7 days, during which the DW was replaced every 4 h. When the liquid outside the dialysis membrane was transparent, the solution was removed from the dialysis membrane and placed in a 4 °C refrigerator for future use.

### 4.4. Characterization of PGC-CDs

TEM (Tecnai G220, FEI Co., Hillsboro, OR, USA) and HR-TEM (JEN-1230, Japan Electron Optics Laboratory, Tokyo, Japan) were used to observe the morphology, particle size distribution and lattice spacing of PGC-CDs [66]. XRD diffractometer (D8-Advanced, Bruker AXS, Karlsruhe, Germany) was performed with Cu K-alpha radiation (wavelength λ = 1.5418 Å) and used to analyze the crystal structure of PGC-CDs. The scanning range was 5–90°, and the scanning rate was 10°/min. Optical properties of 0.5 g/mL of PGC-CDs solution were detected by using fluorescence spectroscopy (F-4500, Tokyo, Japan) and UV-Vis spectrophotometer (CECIL, Cambridge, UK). FTIR spectrometer (Thermo Fisher, Fremont, CA, USA) was used to analyze the surface-group information of the PGC-CDs solution. The spectral resolution was better than 0.25 cm^−1^, the number of scans per sample was 64, and the scanning range was 4000–400 cm^−1^. The data were evaluated using Omnic software. XPS (ESCALAB 250Xi, Thermo Fisher Scientific, Fremont, CA, USA) with a mono X-ray source Al Kα excitation (1486.6 eV) was used to analyze the element composition and surface-active group of PGC-CDs.

### 4.5. Fingerprint Analysis of PGC and PGC-CDs by High-Performance Liquid Chromatography

After pulverizing the PG and PGC-CDs, 200 mg of powder was accurately weighed and added to 10 mL of methanol solution by ultrasonic treatment for 30 min. All solutions were filtered by an 0.22 μm cellulose membrane before HPLC analysis. An Agilent series 1260 HPLC instrument (Agilent Technologies, Waldbronn, Germany) was used to detect PGC and PGC-CDs. The chromatographic column of Ultimate C18 column (250 mm × 4.6 mm, 5 μm) was selected. The mobile phase consisted of acetonitrile (A) and 0.1% phosphoric acid (B). The gradient elution procedure was as follows: 10–20% A at 0–15 min; 20–35% A at 15–50 min; 35–75% A at 50–60 min; 75–80% A at 60–70 min. The flow rate of the mobile phase was 0.5 mL/min, the column temperature was 30 °C, the detection wavelength was 210 nm, and the injection volume was 10 μL.

### 4.6. Cytotoxicity Assessment: CCK-8 Assay

Drug safety is the primary consideration in the development of new drugs. The CCK-8 assay was used to evaluate the cytotoxicity of PGC-CDs on RAW264.7 cells and LO2 cells [67]. RAW264.7 cells in Dulbecco’s modified Eagle’s medium (DMEM) containing 20% fetal bovine serum were cultured in an incubator (37 °C, 5% CO_2_). After the cells were counted, the cells were diluted to 1 × 10^5^ cells/mL, and 100 μL were placed in the control group and PGC-CDs groups of the 96-well plate. Next, 10 mL of phosphate-buffered saline (PBS) was dropped around, and the cells were cultured for 24 h. Then, 100 μL of different concentrations (1000, 500, 250, 125, 62.5, 31.25, 15.62, 7.81 and 3.91 μg/mL) of PGC-CDs solution were added to each well, and the blank group and control group were to join the same amount of DMEM. After the 96-well plate was washed with PBS for 3 times, 10 μL CCK-8 reagent was added to each well for an additional 3 h of incubation. After 3 h, the absorbance of each well at 450 nm was read using a microplate reader. The calculation formula is as follows:$$Cell\;viability\left(\%\right)=\frac{A_{a}-A_{b}}{A_{c}-A_{b}}\times 100$$
where *A_a_*, *A_b_* and *A_c_* represent the absorbance of the PGC-CDs, blank and control groups, respectively.

### 4.7. Evaluation of the Antioxidant Activity of PGC-CDs in Cells

The oxidative stress model of RAW264.7 cells induced by H_2_O_2_ was used to evaluate the antioxidant activity of PGC-CDs in cells [68]. RAW264.7 cells were seeded into a 96-well plate at a seeding density of at 2.0 × 10^5^ cells per well and cultured in an incubator with 5% CO_2_ at 37 °C. Then, 100 μL of different concentrations (0.1, 0.2, 0.3, 0.4, 0.5, 0.6, 0.8 and 1 mmol/L) of H_2_O_2_ solution were added to each well, respectively, and the same amount of DMEM was added to the blank group and the control group. The cytotoxicity of H_2_O_2_ on RAW264.7 cells for 24 h was evaluated by CCK-8 assay, and the optimal concentration of H_2_O_2_-induced oxidative damage was selected. After the above screening, RAW264.7 cells were cultured for 24 h. The treatment groups were pretreated with different concentrations (250, 125 and 62.5 μg/mL) of PGC-CDs for 2 h, and the control group and the model group were given the same amount of DMEM. After 2 h, the model group and PGC-CDs groups were given 0.4 mmol/L H_2_O_2_, and the control group was given the same amount of DMEM for 24 h. After 24 h, the SOD activity and contents of MDA in the cells were detected.

### 4.8. Bilirubin-Induced Hyperbilirubinemia Model and Drug Treatment

Based on previous studies [16,69], this study established the hyperbilirubinemia model through intraperitoneal injection of bilirubin. Seventy mice were randomly divided into 5 groups, with 14 mice in each group: control group, Bil group, and PGC-CDs at high (5.80 mg/kg), medium (2.90 mg/kg) and low (1.45 mg/kg) doses groups. Each group of mice were subjected to intragastric administration of the corresponding drugs once a day for 7 consecutive days. The PGC-CDs groups were given different concentrations of PGC-CDs, while the animals in both the control and Bil groups were given normal saline. One hour after the last intragastric administration, normal saline was intraperitoneally injected into the mice in the control group, and bilirubin solution (150 mg/kg) was intraperitoneally injected into the Bil group and PGC-CDs groups.

### 4.9. General Condition Observation

The general conditions of mice in each group before and after modeling were observed, including spirit, respiration, skin color, hair, excreta and response to external stimulation. Moreover, we recorded the body weight, food intake and number of dead mice in each group before modeling and one week after modeling, and calculated the survival rate.

### 4.10. Neurological Function Observation

According to the Tarlov scores [36,70], the neurological function of mice at different time points before and after modeling was observed and recorded, and the neurological function score was: 0 = no activity; 1 = can move slightly, but cannot stand; 2 = increased activity frequency, but unable to stand on both feet; 3 = can walk a few steps or abnormal gait, can stand; 4 = can walk and stand normally.

### 4.11. Biochemical Analysis

At 24 h after modeling, blood was collected from the orbit of mice and left standing at 4 °C for 4 h; then, the serum was separated by centrifuge (1091× *g*, 10 min). Direct bilirubin (DBIL), indirect bilirubin (IBIL), total bilirubin (TBIL), total bile acid (TBA), alanine aminotransferase (ALT) and aspartate aminotransferase (AST) were determined by the AU-480 automatic biochemical analyzer (Beckman Kurt Co., Ltd., Brea, CA, USA).

### 4.12. Detection of Inflammatory Factors and Markers of Oxidative Stress

After the mice were euthanized, their livers were removed and stored at −80 °C. A total of 300 mg of liver tissues were added to 3 mL PBS (PH 7.4) in an ice bath, and the samples were homogenized with a portable high-speed disperser (Ningbo Xinzhi Co., Ltd., Ningbo, China) for 2 min. The homogenate was centrifuged at 802× *g* for 10 min, and the supernatant was recovered. The levels of IL-6 and TNF-α in serum were detected using ELISA kits, and the levels of SOD (WST-1 method), MDA (TBA method), GSH (Microplate method) and CAT (Visible light) in liver tissue were detected using special kits, according to the manufacturer’s instructions.

### 4.13. Histopathological Analysis

100 mg of liver tissues of mice were excised and fixed with 8 mL of 4% paraformaldehyde solution for more than 48 h. Pathological changes in the liver of mice in each group were observed by H&E staining.

### 4.14. Statistical Analysis

Statistical analysis of experimental data was performed using the Statistical Package for Social Sciences (SPSS, version 20.0). Data results were presented as mean ± standard deviation. Multiple comparisons were performed by one-way ANOVA, followed by the LSD test and Tamhane’s test. *p* < 0.05 indicated that data had statistical difference, *p* < 0.01 was considered a statistically significant difference.

## 5. Conclusions

In summary, PGC-CDs, novel CDs, were prepared by calcinations from PG for the first time. PGC-CDs have been shown to be effective in suppressing levels of inflammatory factors and improving the body’s antioxidant function, demonstrating PGC-CDs may have progressing potential for treating hyperbilirubinemia and preventing liver damage induced by hyperbilirubinemia. This study provided a new idea for research on the material basis of charcoal drugs and a broad prospect for the development of drugs for hyperbilirubinemia and hepatoprotective drugs for hyperbilirubinemia. The present study is a preliminary assessment of the therapeutic effects of PGC-CDs in hyperbilirubinemia. In the future, the distribution of PGC-CDs in vivo after their interaction with bilirubin and their metabolism using fluorescent labeling and other means are needed to explore their implied mechanisms and biological activities. In addition, comparative studies of CDs from different sources in terms of preparation, characterization and pharmacology to find better carbon sources and preparation methods are the focus of our future research, which has important practical significance and application value.

## Figures and Tables

**Figure 1 molecules-28-02720-f001:**
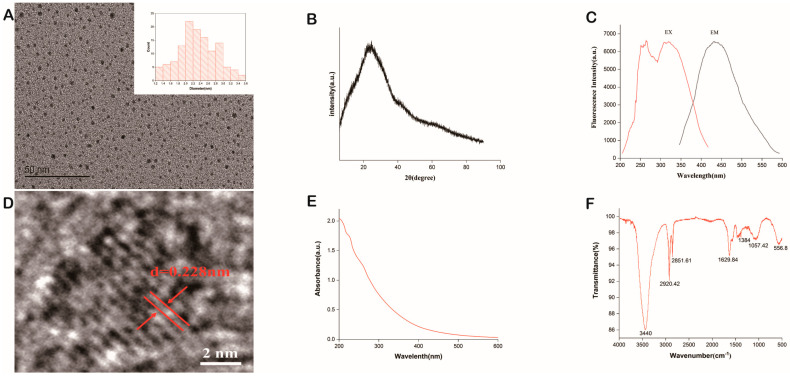
Characterization of *Platycodon grandiflorum*-based carbon dots (PGC-CDs). (**A**) Transmission electron microscope (TEM) images and histogram depicting particle size distribution. (**B**) X-ray Diffraction (XRD) pattern. (**C**) Fluorescence spectrum. (**D**) High-resolution TEM (HRTEM) image. € Ultraviolet-visible (UV-Vis) spectrum. (**F**) Fourier transform infrared (FTIR) spectrum.

**Figure 2 molecules-28-02720-f002:**
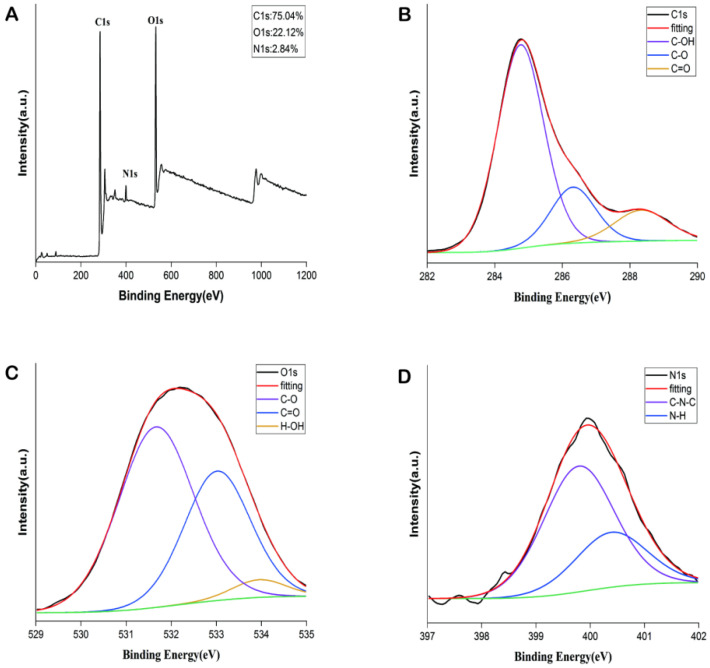
X-ray photoelectron spectroscopy (XPS) spectrum of PGC-CDs. (**A**) Survey spectra. (**B**) C1s. (**C**) O1s. (**D**) N1s.

**Figure 3 molecules-28-02720-f003:**
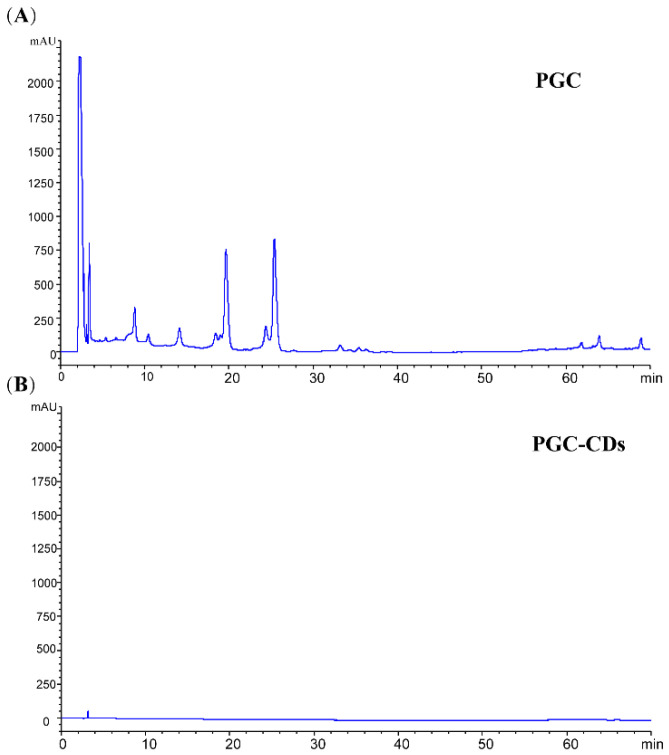
High-performance liquid chromatogram of PGC (**A**) and PGC-CDs (**B**) aqueous solution.

**Figure 4 molecules-28-02720-f004:**
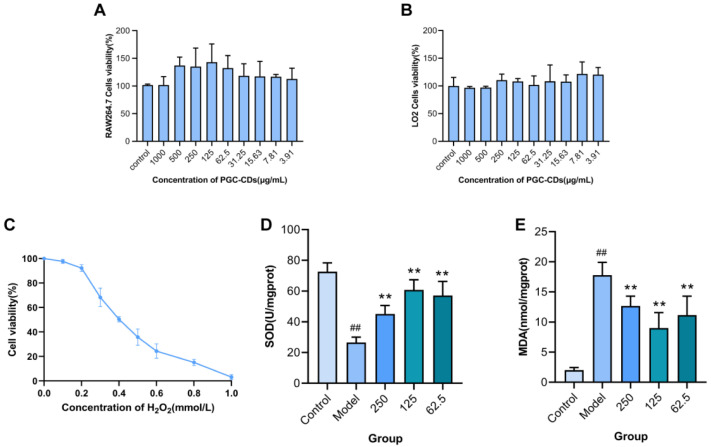
Effects of PGC-CDs on the viability and oxidative stress of cells. (**A**) Effects of various concentrations of PGC-CDs on RAW264.7 cells viability. (**B**) Effects of various concentrations of PGC-CDs on LO2 cells viability. (**C**) Dose-dependent effects of H_2_O_2_ on the viability of RAW264.7 cells. Effects of PGC-CDs on SOD (**D**) and MDA I in RAW264.7 cells with H_2_O_2_-induced model. They were divided into control group (Control), model group (Model), and 250 (250), 125 (125) and 62.5 (62.5) μg/mL of PGC-CDs groups, *n* = 6. Compared with the control group, ^##^ *p* < 0.01. Compared with the model group, ** *p* < 0.01.

**Figure 5 molecules-28-02720-f005:**
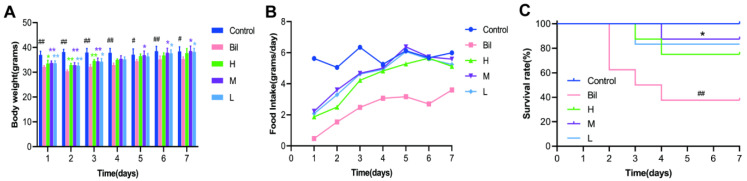
General condition of mice in each group from 1 to 7 days after modeling. (**A**) Body weight; (**B**) Food intake; (**C**) Survival rate. They were divided into control group (Control), Bil group (Bil), and high (H), medium (M) and low (L) dose PGC-CDs groups (5.80, 2.90 and 1.45 mg/kg), *n* = 8. Compared with the control group, ^##^ *p* < 0.01, ^#^ *p* < 0.05. Compared with the Bil group, ** *p* < 0.01, * *p* < 0.05.

**Figure 6 molecules-28-02720-f006:**
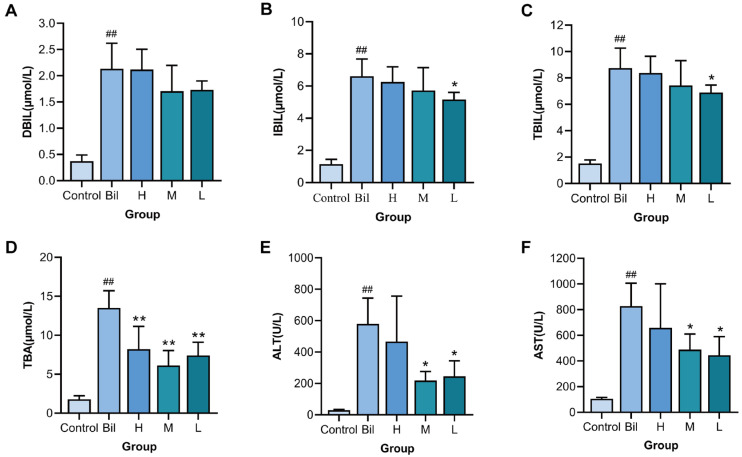
Effects of PGC-CDs on biochemical levels in serum of mice with hyperbilirubinemia model. (**A**) Direct bilirubin (DBIL); (**B**) Indirect bilirubin (IBIL); (**C**) Total bilirubin (TBIL); (**D**)Total bile acid (TBA); I Alanine aminotransferase (ALT); (**F**) Aspartate aminotransferase (AST). They were divided into control group (Control), Bil group (Bil), and high (H), medium (M) and low (L) dose PGC-CDs groups (5.80, 2.90 and 1.45 mg/kg), *n* = 6. Compared with the control group, ^##^ *p* < 0.01. Compared with the Bil group, ** *p* < 0.01, * *p* < 0.05.

**Figure 7 molecules-28-02720-f007:**
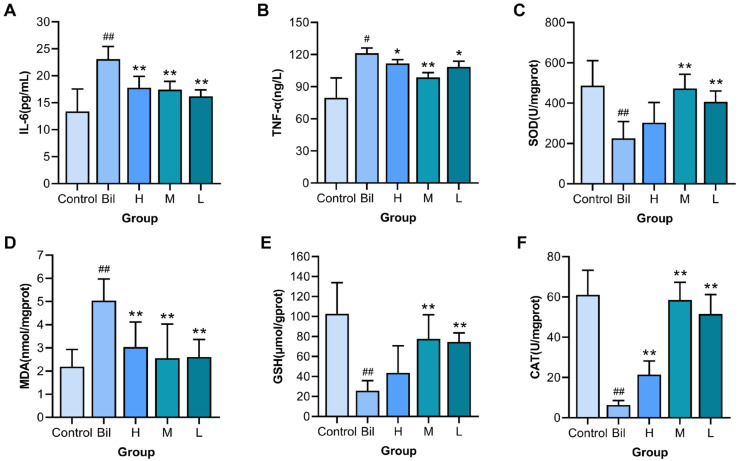
Effects of PGC-CDs on inflammatory factors and oxidative stress in mice. Effects of PGC-CDs on inflammatory factors interleukin-6 (IL-6) (**A**) and tumor necrosis factor-α (TNF-α) (**B**) in serum of mice with hyperbilirubinemia model. Effects of PGC-CDs on superoxide dismutase (SOD) (**C**), malondialdehyde (MDA) (**D**), glutathione (GSH) I and catalase (CAT) (**F**) in liver tissues of mice with hyperbilirubinemia model. They were divided into control group (Control), Bil group (Bil), and high (H), medium (M) and low (L) dose PGC-CDs groups (5.80, 2.90 and 1.45 mg/kg), *n* = 6. Compared with the control group, ^##^ *p* < 0.01, ^#^ *p* < 0.05. Compared with the Bil group, ** *p* < 0.01, * *p* < 0.05.

**Figure 8 molecules-28-02720-f008:**
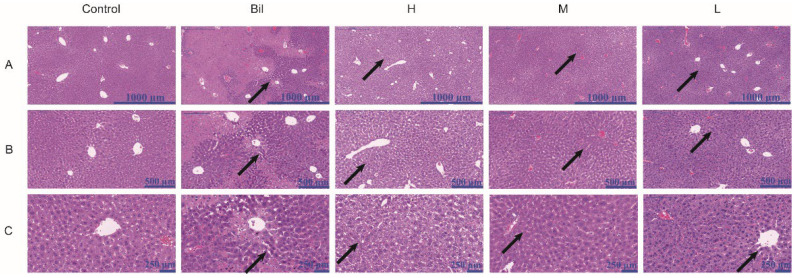
Effects of PGC-CDs on liver pathology in mice. Histological changes of liver obtained from mice of control group (Control), Bil group (Bil), and high (H), medium (M), and low (L) doses of PGC-CDs (5.80, 2.90 and 1.45 mg/kg) in (**A**) magnification at 50×, (**B**) magnification at 100× and (**C**) magnification at 200×.

**Figure 9 molecules-28-02720-f009:**
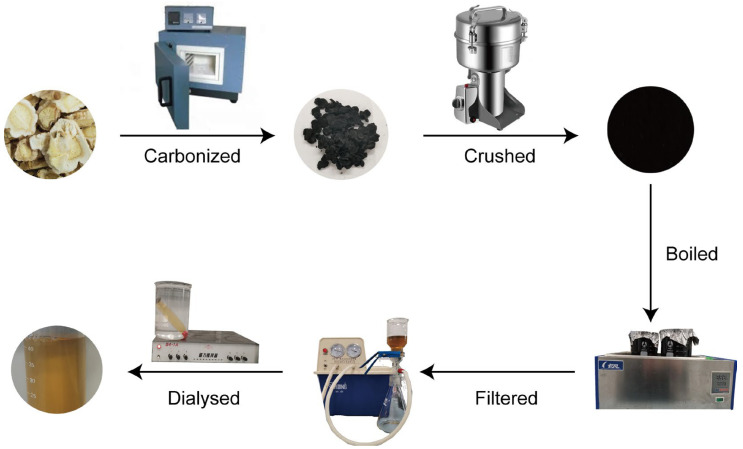
Preparation flow chart of PGC-CDs.

**Table 1 molecules-28-02720-t001:** Tarlov scores of mice in each group at 4 h, 8 h, 12 h, 24 h and 48 h after modeling (Mean ± SD).

Group	Tarlov Score				
	4 h	8 h	12 h	24 h	48 h
Control	4 ± 0	4 ± 0	4 ± 0	4 ± 0	4 ± 0
Bil	1.33 ± 0.52 ^##^	1.17 ± 0.41 ^##^	1.50 ± 0.84 ^##^	1 ± 0 ^##^	1 ± 0 ^##^
H	2.67 ± 0.52 *	2.33 ± 1.03	3.17 ± 0.75 *	2.33 ± 1.21	3.33 ± 1.21 **
M	3.67 ± 0.52 **	3.83 ± 0.41 **	3.67 ± 0.52 **	3.83 ± 0.41 **	3.83 ± 0.41 **
L	3.33 ± 1.21	3.67 ± 0.82 **	3.67 ± 0.82 *	3.17 ± 1.17	3.80 ± 0.45 **

Compared with the control group, ^##^ *p* < 0.01. Compared with the Bil group, ** *p* < 0.01, * *p* < 0.05.

## Data Availability

The data presented in this study are available on request from the corresponding author.

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
