# Peer review of "A Novel Drug with Potential to Treat Hyperbilirubinemia and Prevent Liver Damage Induced by Hyperbilirubinemia: Carbon Dots Derived from Platycodon grandiflorum"

_molecules, 2023, doi:10.3390/molecules28062720_

Round 1
Reviewer 1 Report (Previous Reviewer 3)
ID: molecules-2253487
Title: A novel drug with potential to treat hyperbilirubinemia and prevent liver damage induced by hyperbilirubinemia: Carbon dots derived from Platycodon grandiflorum
The authors described the synthesis and characterization of carbon dots from Platycodon grandiflorum. The obtained nanoparticles were tested in the treatment of hyperbilirubinemia. In my opinion, the characterization of nanoparticles can be improved with some missing information.
I have the following comments:
1. The authors should highlight the differences/similarities between PGC-CDs and other CDs obtained from plants.
2. “FTIR spectroscopy was used to analyze the surface group information in PGC-CDs solution” – If the FTIR spectrum is obtained for a solution, the bands assigned to solvent (water) must be highlighted. A comparison with the H2O spectrum can be useful.
3. Some expressions need to be revised, e.g. “a novel CDs”, “a novel carbon dots”, “Ultraviolet–visible (UV-Vis) of PGC-CDs” (spectrum?), “small molecule compounds were basically pyrolyzed”, “The optical characteristics of PGC-CDs solution was analyzed by fluorescence spectroscopy and UV-Vis spectrophotometer.”
4. The names of spectrometers (fluorescence, UV-vis, FTIR), XRD diffractometer and TEM microscope are missing.
5. A mention about the method for determination of MDA, SOD, GSH and CAT is necessary.
6. “XPS was used to analyze the element composition and bonding actions of PGC-CDs.” – What means more accurate “bonding actions”?
7. “high-temperature calcination” – The calcination is a thermal treatment, so it is a pleonasm. Furthermore, a temperature of 350 ℃ is not a “high-temperature” for calcination.
Author Response
Please see the attachment.

Reviewer 2 Report (New Reviewer)
The manuscript "A novel drug with potential to treat hyperbilirubinemia and prevent liver damage induced by hyperbilirubinemia: Carbon
dots derived from Platycodon grandiflorum" may be of interest to the readers of this journal. However, I have some comments that I have attached to the manuscript, which the authors need to use to improve the manuscript. Thanks

Author Response
Please see the attachment.

Reviewer 3 Report (New Reviewer)
Line 106: 2θ = 22.765â—¦ is not understood, correct.
Line 117: correct CH2, it should be the subscript number.
Line 128: 284.76 eV, 531.83 eV and 399.36 eV, can be represented as follows: 284.76, 531.83 and 399.36 eV, like lines 130, 133, review throughout the manuscript.
Line 205: Separate units of values 2.90, 1.45mg/kg, so it should be 2.90, 1.45 mg/kg, review the entire manuscript and change
Line 431, the degree symbol is underlined, it shouldn't be. You must homogenize, because in other data the symbol is observed smaller.
Line 445: Unite g/ mL, it should look like this, g/mL homogenize throughout the manuscript
Line 466: 75%-80% must remain; 75-80%. Homogenize throughout the manuscript
Line 467: 0.5 mL/min, is separate, should be 0.5 mL/min
Line 467: It's too far apart 30°C, it should be 30°C.
Line 477: should be …7.81, and 3.91 µg/mL
Line 526 and 531: the r/min, convert them to “G”
Citations must be in [20] and in normal form and not as superscript. Review writing references: https://www.mdpi.com/journal/molecules/instructions
Line 495 is very separated from the comma of 125, 62.5, also add "and"
In all links replace the C-C symbol with this other C‒C
In several parts of the manuscript Fourier transform infrared (FTIR) is observed, it should only be mentioned like this once and then FTIR, also review XPS and others to change.
Homogenize the figures, for example, the X axis of figures 2B and 2C, they intersect at 290 and 534 respectively, they must end the same as figures 2A and 2D at 1200 and 402 eV respectively, so that they are all homogenized, check in the other figures.
In Figure 4, it appears that Figure A and B are not the same size.
The probability symbol p< must be lowercase and not uppercase. Review and change throughout the manuscript
Figures 5A and 5B are not sharp enough. If possible leave her bigger by herself.
In figure 8. You must add a line that has a known size, that is; 1000 µm to be able to understand the real size of what is indicated.
Round 2
Reviewer 2 Report (New Reviewer)
Thanks for improving the manuscript. This is absolutely a better manuscript than the previous one.I recommend the manuscript to be accepted
This manuscript is a resubmission of an earlier submission. The following is a list of the peer review reports and author responses from that submission.
Round 1
Reviewer 1 Report
In this manuscript, the authors prepared the Platycodon grandiflorum (PG)-based carbon dots. This PGC-CDs may have progressing potentials in treating hyperbilirubinemia and preventing liver damage induced by hyperbilirubinemia. But there are some important issues should be taken in consideration to improve this manuscript (Major revision).
1. In this study, the authors had to use normal hepatocytes to verify cytotoxicity.
2. What is the basis for the high (5.80 mg/kg), medium (2.90 mg/kg) and low (1.45 mg/kg) dose settings?
3. What is the relationship between nerve injury and hyperbilirubinemia? The author does not fully understand the causal relationship between the two.
4. The “Tarlov scores” does not seem to apply to hyperbilirubinemia.
5. In the “2.7. Effects of PGC-CDs on inflammatory factors” section, “...that liver damage induced by hyperbilirubinemia may be related to inflammation” is wrong. The authors' available data do not support this conclusion.
6. In Figure 7, the size of the image is clearly inconsistent and the author must add the scale. And the authors need to mark abnormal pathological areas in the diagram.
7. After dialysis, whether the substance of dialysis has therapeutic effect? The authors need to add experiments to rule out this possibility.
8. In the Discussion, the difference between PG Carbonisata and PGC-CDs needs to be elaborated.
9. Some recent published papers about herbal medicine-CDs are suggested to be referred in this manuscript: Nano Research. 15, 9274–9285 (2022). Nano Research. 15, 1699–1708 (2022). Journal of Nanobiotechnology 19, 320 (2021).
10. The manuscript has a considerable amount of unprofessional boasting that needs to be toned.
Author Response
Responses to the reviewer:
Comments and Suggestions for Authors
In this manuscript, the authors prepared the Platycodon grandiflorum (PG)-based carbon dots. This PGC-CDs may have progressing potentials in treating hyperbilirubinemia and preventing liver damage induced by hyperbilirubinemia. But there are some important issues should be taken in consideration to improve this manuscript (Major revision).
Response: Thank you for your comments and suggestion concerning our manuscript. The comments and suggestions are all valuable and very helpful for revising and improving our paper, as well as the important guiding significance to our research. We have studied comments carefully and have made correct on which we hope meet with approval.
- In this study, the authors had to use normal hepatocytes to verify cytotoxicity.
Response: In order to take the reviewer concern into account, and improve the quality of our manuscript, the cytotoxicity of PGC-CDs in hepatocytes is evaluated in the revised manuscript. The specific content is described below:
As shown in Supplementary Figure S1, the cell viability of LO2 cells co-cultured with PGC-CDs (3.91–250 µg/mL) was more than 100%, indicating that PGC-CDs did not inhibit cell growth and even promoted cell proliferation. However, when the concentration of PGC-CDs increased from 500 to 1000 µg/mL, the cell viability of LO2 cells began to decrease slightly. We can conclude that PGC-CDs did not affect the cell viability of LO2 cells at the concentration from 1000 to 3.91 μg/mL (Please see the revised manuscript, line 147-148).
Figure S1. Effects of various concentrations of PGC-CDs on LO2 cells viability.
- What is the basis for the high (5.80 mg/kg), medium (2.90 mg/kg) and low (1.45 mg/kg) dose settings?
Response: Thanks for your excellent question. In order to ensure the functional bioactivities with low biotoxicity, the concentration mentioned of PGC-CDs in this article was inspired by many published studies about biomass-based carbon dots [1-3]. Meanwhile, the safety dose of PGC-CDs was evaluated by RAW264.7 cells ranging from 1000 to 3.91 μg/mL, which result showed that almost dose of PGC-CDs had low cytotoxicity.
References
- Zhang, Y.; Wang, S.; Lu, F.; Zhang, M.; Kong, H.; Cheng, J.; Luo, J.; Zhao, Y.; Qu, H., The neuroprotective effect of pretreatment with carbon dots from Crinis Carbonisatus (carbonized human hair) against cerebral ischemia reperfusion injury. Journal of nanobiotechnology 2021, 19 (1), 257.
- Wang, C.; Ji, Y.; Cao, X.; Yue, L.; Chen, F.; Li, J.; Yang, H.; Wang, Z.; Xing, B., Carbon Dots Improve Nitrogen Bioavailability to Promote the Growth and Nutritional Quality of Soybeans under Drought Stress. ACS nano 2022, 16 (8), 12415-12424.
- Lu, F.; Ma, Y.; Huang, H.; Zhang, Y.; Kong, H.; Zhao, Y.; Qu, H.; Wang, Q.; Liu, Y.; Kang, Z., Edible and highly biocompatible nanodots from natural plants for the treatment of stress gastric ulcers. Nanoscale 2021, 13 (14), 6809-6818.
- What is the relationship between nerve injury and hyperbilirubinemia? The author does not fully understand the causal relationship between the two.
Response: It is a valid point that the relationship between nerve injury and hyperbilirubinemia.
Hyperbilirubinemia is a global disease that often occurs in newborns. If it is not intervened in time, when the metabolism and elimination of bilirubin in the body is imbalance, it will form neurotoxic bilirubinemia, which can cause neonatal nervous system damage, or even irreversible nervous system damage, even cause death [1-4]. When bilirubin exceeds the binding capacity of serum protein, free bilirubin quickly passes through the blood-brain barrier and deposits in important brain nuclei such as the basal ganglia, hippocampus, brainstem, cerebellum, cochlear nucleus, medulla oblongata of the medial thalamus, etc. It will cause nerve damage. In severe cases, it may cause functional damage to specific central nuclei [5-8], which may cause sequelae such as dyskinesia, dystonia [9], mental retardation [10,11], visual impairment [9], and hearing loss [12,13], even language retardation, autism, low learning and memory ability are all related to neonatal hyperbilirubinemia [14-19].
The pathogenesis of hyperbilirubinemia is complex. Bilirubin acts on neurovascular endothelial cells, affecting the development of vascular endothelial growth factor, reducing the closure and connection between nerve cells, and increasing the resistance and permeability between endothelial cells, making bilirubin more likely to penetrate the blood-brain barrier and affect the function of nerve cells. The exposure of neurons to high levels of bilirubin can cause neuronal atrophy, cell apoptosis, resulting in reduced branching and growth inhibition, thus affecting the development and maturation of nerve cells. The changes in important components of neurons, axons, synapses, dendrites and so on can lead to the apoptosis of nerve cells of important brain structures. Long-term exposure to high bilirubin level can increase the autophagy of apoptotic nerve cells in the body, gradually decrease the function of surviving nerve cells, and affect nerve conduction function [20,21].
The neurotoxicity of bilirubin is mainly related to the concentration of free bilirubin in serum. High concentration of free bilirubin can easily pass through the blood-brain barrier and enter the brain, causing excitotoxicity, promoting the release of free radicals and pro-inflammatory factors, leading to plasma membrane disturbance, oxidative stress, neuritis and other pathological changes, making nerve cells denature and apoptosis, resulting in nerve dysfunction eventually [22-24]. Ostrow et al. [25,26] found that slightly elevated free bilirubin has neurotoxicity effect. Studies [27] have shown that the accumulation of free bilirubin in cerebrospinal fluid and central nervous system may be related to factors such as the active uptake of bilirubin by the blood-brain barrier and cell membrane as well as the influence of external pumping bilirubin on its neurotoxicity. In addition, free bilirubin not only has toxic effects on astrocytes and neurons, but also affects the excitability of neurons in the nervous system at a low concentration and interfere with the transmission of information between synapses of neurons. At the same time, it can also damage mitochondria, leading to cell energy metabolism disorders and cell apoptosis [25,26,28].
The possible mechanism of bilirubin-induced apoptosis of nerve cells involves multiple pathways such as intracellular calcium overload and mitochondrial damage to promote cell apoptosis or necrosis [29-32]. In addition, excitatory amino acids participate in the process of apoptosis [33]. Unconjugated bilirubin (UCB) can significantly inhibit the uptake of glutamate by astrocytes. It can be seen that the increase of excitatory amino acids may lead to cell necrosis and apoptosis through the activation of NMDA (N-methyl-D-aspartate) receptor [34]. In addition, UCB can significantly inhibit the synaptic transmission of hippocampal neurons, reduce the activity of protein kinase C and calmodulin, and induce delayed cell death [35]. Although considerable progress has been made in the study of bilirubin neurotoxicity, the mechanism of selective damage of bilirubin to specific nuclei and the long-term sequelae of bilirubin encephalopathy need to be revealed.
References
- Cohen, R. S.; Wong, R. J.; Stevenson, D. K., Understanding neonatal jaundice: a perspective on causation. Pediatrics and neonatology 2010, 51 (3), 143-8.
- Du, L.; Ma, X.; Shen, X.; Bao, Y.; Chen, L.; Bhutani, V. K., Neonatal hyperbilirubinemia management: Clinical assessment of bilirubin production. Seminars in perinatology 2021, 45 (1), 151351.
- Kumral, A.; Ozkan, H.; Duman, N.; Yesilirmak, D. C.; Islekel, H.; Ozalp, Y., Breast milk jaundice correlates with high levels of epidermal growth factor. Pediatric research 2009, 66 (2), 218-21.
- He, C. H.; Qu, Y., [Research advances in neonatal hyperbilirubinemia and gene polymorphisms]. Zhongguo dang dai er ke za zhi = Chinese journal of contemporary pediatrics 2020, 22 (3), 280-284.
- Wennberg, R. P.; Ahlfors, C. E.; Bhutani, V. K.; Johnson, L. H.; Shapiro, S. M., Toward understanding kernicterus: a challenge to improve the management of jaundiced newborns. Pediatrics 2006, 117 (2), 474-85.
- Fujiwara, R.; Nguyen, N.; Chen, S.; Tukey, R. H., Developmental hyperbilirubinemia and CNS toxicity in mice humanized with the UDP glucuronosyltransferase 1 (UGT1) locus. Proceedings of the National Academy of Sciences of the United States of America 2010, 107 (11), 5024-9.
- Ostrow, J. D.; Tiribelli, C., Bilirubin induced neurological damage: from the cell to the newborn. Current pharmaceutical design 2009, 15 (25), 2868.
- Watchko, J. F.; Tiribelli, C., Bilirubin-induced neurologic damage--mechanisms and management approaches. The New England journal of medicine 2013, 369 (21), 2021-30.
- Paludetto, R.; Mansi, G.; Raimondi, F.; Romano, A.; Crivaro, V.; Bussi, M.; D'Ambrosio, G., Moderate hyperbilirubinemia induces a transient alteration of neonatal behavior. Pediatrics 2002, 110 (4), e50.
- Newman, T. B.; Maisels, M. J., Evaluation and treatment of jaundice in the term newborn: a kinder, gentler approach. Pediatrics 1992, 89 (5 Pt 1), 809-18.
- Arun Babu, T.; Bhat, B. V.; Joseph, N. M., Association between peak serum bilirubin and neurodevelopmental outcomes in term babies with hyperbilirubinemia. Indian journal of pediatrics 2012, 79 (2), 202-6.
- Ostrow, J. D.; Pascolo, L.; Shapiro, S. M.; Tiribelli, C., New concepts in bilirubin encephalopathy. European journal of clinical investigation 2003, 33 (11), 988-97.
- Oysu, C.; Aslan, I.; Ulubil, A.; Baserer, N., Incidence of cochlear involvement in hyperbilirubinemic deafness. The Annals of otology, rhinology, and laryngology 2002, 111 (11), 1021-5.
- Vianello, E.; Zampieri, S.; Marcuzzo, T.; Tordini, F.; Bottin, C.; Dardis, A.; Zanconati, F.; Tiribelli, C.; Gazzin, S., Histone acetylation as a new mechanism for bilirubin-induced encephalopathy in the Gunn rat. Scientific reports 2018, 8 (1), 13690.
- Vodret, S.; Bortolussi, G.; Jašprová, J.; Vitek, L.; Muro, A. F., Inflammatory signature of cerebellar neurodegeneration during neonatal hyperbilirubinemia in Ugt1 (-/-) mouse model. Journal of neuroinflammation 2017, 14 (1), 64.
- Maimburg, R. D.; Bech, B. H.; Vaeth, M.; Møller-Madsen, B.; Olsen, J., Neonatal jaundice, autism, and other disorders of psychological development. Pediatrics 2010, 126 (5), 872-8.
- Mukhopadhyay, K.; Chowdhary, G.; Singh, P.; Kumar, P.; Narang, A., Neurodevelopmental outcome of acute bilirubin encephalopathy. Journal of tropical pediatrics 2010, 56 (5), 333-6.
- Shapiro, S. M., Chronic bilirubin encephalopathy: diagnosis and outcome. Seminars in fetal & neonatal medicine 2010, 15 (3), 157-63.
- Zhang, L.; Liu, W.; Tanswell, A. K.; Luo, X., The effects of bilirubin on evoked potentials and long-term potentiation in rat hippocampus in vivo. Pediatric research 2003, 53 (6), 939-44.
- Palmela, I.; Sasaki, H.; Cardoso, F. L.; Moutinho, M.; Kim, K. S.; Brites, D.; Brito, M. A., Time-dependent dual effects of high levels of unconjugated bilirubin on the human blood-brain barrier lining. Frontiers in cellular neuroscience 2012, 6, 22.
- Brito, M. A.; Zurolo, E.; Pereira, P.; Barroso, C.; Aronica, E.; Brites, D., Cerebellar axon/myelin loss, angiogenic sprouting, and neuronal increase of vascular endothelial growth factor in a preterm infant with kernicterus. Journal of child neurology 2012, 27 (5), 615-24.
- Kato, S.; Iwata, O.; Yamada, Y.; Kakita, H.; Yamada, T.; Nakashima, H.; Sugiura, T.; Suzuki, S.; Togari, H., Standardization of phototherapy for neonatal hyperbilirubinemia using multiple-wavelength irradiance integration. Pediatrics and neonatology 2020, 61 (1), 100-105.
- Kanmaz, H. G.; Okur, N.; Dilli, D.; YeÅŸilyurt, A.; OÄŸuz Åž, S., The effect of phototherapy on sister chromatid exchange with different light density in newborn hyperbilirubinemia. Turk pediatri arsivi 2017, 52 (4), 202-207.
- Amin, S. B.; Smith, T.; Timler, G., Developmental influence of unconjugated hyperbilirubinemia and neurobehavioral disorders. Pediatric research 2019, 85 (2), 191-197.
- Orozco-Ibarra, M.; Estrada-Sánchez, A. M.; Massieu, L.; Pedraza-Chaverrí, J., Heme oxygenase-1 induction prevents neuronal damage triggered during mitochondrial inhibition: role of CO and bilirubin. The international journal of biochemistry & cell biology 2009, 41 (6), 1304-14.
- Ostrow, J. D.; Pascolo, L.; Tiribelli, C., Reassessment of the unbound concentrations of unconjugated bilirubin in relation to neurotoxicity in vitro. Pediatric research 2003, 54 (1), 98-104.
- Leibinger, M.; Müller, A.; Andreadaki, A.; Hauk, T. G.; Kirsch, M.; Fischer, D., Neuroprotective and axon growth-promoting effects following inflammatory stimulation on mature retinal ganglion cells in mice depend on ciliary neurotrophic factor and leukemia inhibitory factor. The Journal of neuroscience : the official journal of the Society for Neuroscience 2009, 29 (45), 14334-41.
- Fernandes, A.; Falcão, A. S.; Silva, R. F.; Gordo, A. C.; Gama, M. J.; Brito, M. A.; Brites, D., Inflammatory signalling pathways involved in astroglial activation by unconjugated bilirubin. Journal of neurochemistry 2006, 96 (6), 1667-79.
- Schjott, J. M.; Plummer, M. R., Sustained activation of hippocampal Lp-type voltage-gated calcium channels by tetanic stimulation. The Journal of neuroscience : the official journal of the Society for Neuroscience 2000, 20 (13), 4786-97.
- Liang, M.; Yin, X. L.; Shi, H. B.; Li, C. Y.; Li, X. Y.; Song, N. Y.; Shi, H. S.; Zhao, Y.; Wang, L. Y.; Yin, S. K., Bilirubin augments Ca(2+) load of developing bushy neurons by targeting specific subtype of voltage-gated calcium channels. Scientific reports 2017, 7 (1), 431.
- Rodrigues, C. M.; Solá, S.; Silva, R.; Brites, D., Bilirubin and amyloid-beta peptide induce cytochrome c release through mitochondrial membrane permeabilization. Molecular medicine (Cambridge, Mass.) 2000, 6 (11), 936-46.
- Kirsch, D. G.; Doseff, A.; Chau, B. N.; Lim, D. S.; de Souza-Pinto, N. C.; Hansford, R.; Kastan, M. B.; Lazebnik, Y. A.; Hardwick, J. M., Caspase-3-dependent cleavage of Bcl-2 promotes release of cytochrome c. The Journal of biological chemistry 1999, 274 (30), 21155-61.
- Grojean, S.; Koziel, V.; Vert, P.; Daval, J. L., Bilirubin induces apoptosis via activation of NMDA receptors in developing rat brain neurons. Experimental neurology 2000, 166 (2), 334-41.
- Silva, R.; Mata, L. R.; Gulbenkian, S.; Brito, M. A.; Tiribelli, C.; Brites, D., Inhibition of glutamate uptake by unconjugated bilirubin in cultured cortical rat astrocytes: role of concentration and pH. Biochemical and biophysical research communications 1999, 265 (1), 67-72.
- Churn, S. B.; DeLorenzo, R. J.; Shapiro, S. M., Bilirubin induces a calcium-dependent inhibition of multifunctional Ca2+/calmodulin-dependent kinase II activity in vitro. Pediatric research 1995, 38 (6), 949-54.
- The “Tarlov scores” does not seem to apply to hyperbilirubinemia.
Response: We thank the reviewer for asking for the application of “Tarlov scores” to hyperbilirubinemia. Severe hyperbilirubinemia causes severe neurological dysfunction [1], leading to acute bilirubin encephalopathy (ABE). Bilirubin encephalopathy is often divided into acute bilirubin encephalopathy and chronic postkernicteric bilirubin encephalopathy according to the nervous system toxicity. Acute bilirubin encephalopathy can be divided into three stages according to clinical manifestations: the first stage is characterized by hypotonia and dyspnea; In the second stage, hypotonia turns into hypertonia, which is manifested by angular tension, neck stiffness, irritability, irritability, fever, etc. With the further damage of bilirubinemia, the third stage is manifested by epilepsy, coma, apnea, and even death [2]. Johnson believed that the diagnostic criteria for acute bilirubin encephalopathy should have at least three manifestations, namely irritation, increased muscle tension, early head backward tilt and opisthotonus [3]. Chronic postkernicteric bilirubin encephalopathy is mainly characterized by extrapyramidal motor abnormalities, fixation abnormalities, hearing impairment and intellectual impairment. This description is consistent with the program of the American Academy of Pediatrics in 2004 [4]. In addition, in a long-term follow-up study, it was found that the symptoms of hypotonia and dystonia in children with nervous dysfunction caused by hyperbilirubinemia were still significantly higher than those in the normal group [5]. In conclusion, nerve injury caused by hyperbilirubinemia may cause movement disorders such as dystonia in clinical manifestations.
Tarlov scoring is the motion evaluation criteria proposed by Tarlov in 1953 for animals after spinal compression injury. It has been widely used after being improved by scholars [6]. In many studies, the Tarlov score has been used to evaluate the neurological and motor function of animals according to the observation of the motor status of the limbs, such as the evaluation of the angiogenesis ability of atherosclerotic rats [7], the delayed neural functional defects after the repair of thoracoabdominal aortic aneurysm [8], the ischemic stroke [9], the degree of nerve damage caused by hyperbilirubinemia [10], etc. The results of this experiment showed that compared with the control group, the neurobehavioral scores of mice in the Bil group were significantly decreased, and the difference was statistically significant (P<0.01). Therefore, the improved Tarlov score can be used to preliminarily evaluate the neurological function of hyperbilirubinemia mice, so as to increase the understanding of the disease and guide the treatment.
References
- Feng, J.; Li, M.; Wei, Q.; Li, S.; Song, S.; Hua, Z., Unconjugated bilirubin induces pyroptosis in cultured rat cortical astrocytes. Journal of neuroinflammation 2018, 15 (1), 23.
- Sgro, M.; Campbell, D.; Barozzino, T.; Shah, V., Acute neurological findings in a national cohort of neonates with severe neonatal hyperbilirubinemia. Journal of perinatology : official journal of the California Perinatal Association 2011, 31 (6), 392-6.
- Johnson LH. Reply:. J Pediatr 2003, 141, 214-215.
- Shapiro, S. M., Definition of the clinical spectrum of kernicterus and bilirubin-induced neurologic dysfunction (BIND). Journal of perinatology : official journal of the California Perinatal Association 2005, 25 (1), 54-9.
- Soorani-Lunsing, I.; Woltil, H. A.; Hadders-Algra, M., Are moderate degrees of hyperbilirubinemia in healthy term neonates really safe for the brain? Pediatric research 2001, 50 (6), 701-5.
- Basso, D. M.; Beattie, M. S.; Bresnahan, J. C., A sensitive and reliable locomotor rating scale for open field testing in rats. Journal of neurotrauma 1995, 12 (1), 1-21.
- Zeng, J.; Lu, C.; Huang, H.; Huang, J., Effect of Recombinant Netrin-1 Protein Combined with Peripheral Blood Mesenchymal Stem Cells on Angiogenesis in Rats with Arteriosclerosis Obliterans. BioMed research international 2022, 2022, 3361605.
- Safi, H. J.; Miller, C. C., 3rd; Azizzadeh, A.; Iliopoulos, D. C., Observations on delayed neurologic deficit after thoracoabdominal aortic aneurysm repair. Journal of vascular surgery 1997, 26 (4), 616-22.
- Chen L.; Duan X.L.; Zhang Y.S.; Ma X.Y.; Guo Y.; Jiang N.M., Effect of Zhuang medicine medicinal thread moxibustion combined with electroacupuncture therapy for rats having ischemic stroke with flaccid paralysis. Guangxi medical journal 2020, 42(18), 2400-2403.
- Zhang S.B.; Zhou X.Y.; Gong C.L.;, Effects of naloxone on Bcl-2/Bax pathway and neurological dysfunction in neonatal rats with hyperbilirubinemia. Chinese journal of birth health and heredity 2021,29(10), 1378-1382.
- In the “2.7. Effects of PGC-CDs on inflammatory factors” section, “...that liver damage induced by hyperbilirubinemia may be related to inflammation” is wrong. The authors' available data do not support this conclusion.
Response: Thank you for the reviewer’s question. The production and release of inflammatory factors such as IL-6 and TNF-α are the result of external stimulatory signals mediated to the nucleus through various signal transduction pathways in cells. In this study, the levels of IL-6 and TNF-α in the Bil group were significantly higher than those in the control group (P<0.05), indicating that liver damage induced by hyperbilirubinemia may stimulate the production of inflammatory factors, which was consistent with the literature reports that hyperbilirubinemia could aggravate inflammation by activating the NF-κB signaling pathway, thus causing inflammation [1-4]. NF-κB, as an inflammatory transcription factor, regulates the expression of TNF-α, COX-2, IL-1β and other inflammatory factor genes [5,6]. IL-6 and TNF-α are inflammatory factors synthesized and secreted by mononuclear macrophages, lymphocytes and other inflammatory cells. IL-6 can stimulate T cells, activate B cells, and stimulate hepatocytes to synthesize hepatocyte acute reactive proteins, such as C-reactive protein. TNF-α mediates white blood cells to adhere to vascular endothelial cells, activates inflammatory cells to kill microorganisms, activates T cells, stimulates B cells to produce antibodies, and stimulates monocytes to produce other cytokines. In addition, studies [7,8] have shown that IL-6H and TNF-α are involved in liver damage, and are closely related to liver cell injury and disease severity. Therefore, we speculate that liver damage caused by hyperbilirubinemia may be related to inflammatory, but further studies are needed to confirm this idea. We have revised the description as follows: indicating that liver damage induced by hyperbilirubinemia may stimulate the production of inflammatory factors (Please see the revised manuscript, line 255-256).
References
- Sun, X. M.; Kang, P.; Tao, K., Causes of immune dysfunction in hyperbilirubinemia model rats. Asian Pacific journal of tropical medicine 2015, 8 (5), 382-5.
- Song, S.; Zhu, Y.; Dang, S.; Wang, S.; Hua, Z., [Role of nuclear factor-κB activation in bilirubin-induced rat hippocampal neuronal apoptosis and the effect of TAT-NBD intervention]. Nan fang yi ke da xue xue bao = Journal of Southern Medical University 2013, 33 (2), 172-6.
- Fernandes, A.; Barateiro, A.; Falcão, A. S.; Silva, S. L.; Vaz, A. R.; Brito, M. A.; Silva, R. F.; Brites, D., Astrocyte reactivity to unconjugated bilirubin requires TNF-α and IL-1β receptor signaling pathways. Glia 2011, 59 (1), 14-25.
- Mazzone, G. L.; Rigato, I.; Ostrow, J. D.; Tiribelli, C., Bilirubin effect on endothelial adhesion molecules expression is mediated by the NF-kappaB signaling pathway. Bioscience trends 2009, 3 (4), 151-7.
- He, X.; Shu, J.; Xu, L.; Lu, C.; Lu, A., Inhibitory effect of Astragalus polysaccharides on lipopolysaccharide-induced TNF-a and IL-1β production in THP-1 cells. Molecules (Basel, Switzerland) 2012, 17 (3), 3155-64.
- Min, S. Y.; Hwang, S. Y.; Jung, Y. O.; Jeong, J.; Park, S. H.; Cho, C. S.; Kim, H. Y.; Kim, W. U., Increase of cyclooxygenase-2 expression by interleukin 15 in rheumatoid synoviocytes. The Journal of rheumatology 2004, 31 (5), 875-83.
- Yeo, W.; Mo, F. K.; Chan, S. L.; Leung, N. W.; Hui, P.; Lam, W. Y.; Mok, T. S.; Lam, K. C.; Ho, W. M.; Koh, J.; Tang, J. W.; Chan, A. T.; Chan, P. K., Hepatitis B viral load predicts survival of HCC patients undergoing systemic chemotherapy. Hepatology (Baltimore, Md.) 2007, 45 (6), 1382-9.
- Mordes, D. A.; Brachtel, E. F., Cytopathology of subacute thyroiditis. Diagnostic cytopathology 2012, 40 (5), 433-4.
- In Figure 7, the size of the image is clearly inconsistent and the author must add the scale. And the authors need to mark abnormal pathological areas in the diagram.
Response: We sincerely appreciate the reviewer for careful reading, and apologize for the error in Figure 7. We have adjusted the image size and added the scale. At the same time, abnormal pathological areas are marked in the figure. The revised figure 7 can be found in the revised manuscript (Please see the revised manuscript, line 310). The revised Figure 7 is shown below:
Figure 7. Effects of PGC-CDs on liver pathology in mice. Histological changes of liver obtained from mice of control group (Control), Bil group (Bil) and high (H), medium (M), and low (L) doses of PGC-CDs (5.80, 2.90, 1.45mg/kg) in (A) magnification at 50X, (B) magnification at 100X and (C) magnification at 200X.
- After dialysis, whether the substance of dialysis has therapeutic effect? The authors need to add experiments to rule out this possibility.
Response: Thank you very much for important comments. The reviewer’s comments are really thoughtful. At present, there is no report on the treatment of hyperbilirubinemia with the substances of Platycodon grandiflorum Carbonisata (PGC) dialysis. The dialysis membrane is a semi-permeable membrane and usually used to separation of specific molecular weight components. In our previous study, we have reported that the outside solution of dialysis bag remained the small molecular components and related bioactivity of the precursor. In contrast, the inside solution evaluated by HPLC have no obvious peak related to the small molecular components, as well as shows new pharmacological effects [1,2]. Inspired by ancient Chinese medicine books, we found that the carbon dots in PGC had this effect. In addition, we have removed the small and medium molecular active ingredients of PGC through dialysis during the preparation, and PGC-CDs were new active ingredients. The study found that the J-CDs extracted from red jujube almost contained no Fe3+ in red jujube and had hematopoietic capacity [3]. Of course, the idea is very interesting. We will compare the biological effects between PGC-CDs and dialyzed substances while further studying the biological activity of PG.
References
- Zhao, Y.; Zhang, Y.; Kong, H.; Cheng, G.; Qu, H.; Zhao, Y., Protective Effects of Carbon Dots Derived from Armeniacae Semen Amarum Carbonisata Against Acute Lung Injury Induced by Lipopolysaccharides in Rats. International journal of nanomedicine 2022, 17, 1-14.
- Liu, X.; Wang, Y.; Yan, X.; Zhang, M.; Zhang, Y.; Cheng, J.; Lu, F.; Qu, H.; Wang, Q.; Zhao, Y., Novel Phellodendri Cortex (Huang Bo)-derived carbon dots and their hemostatic effect. Nanomedicine (London, England) 2018, 13 (4), 391-405.
- Xu, Y.; Wang, B.; Zhang, M.; Zhang, J.; Li, Y.; Jia, P.; Zhang, H.; Duan, L.; Li, Y.; Li, Y.; Qu, X.; Wang, S.; Liu, D.; Zhou, W.; Zhao, H.; Zhang, H.; Chen, L.; An, X.; Lu, S.; Zhang, S., Carbon Dots as a Potential Therapeutic Agent for the Treatment of Cancer-Related Anemia. Advanced materials (Deerfield Beach, Fla.) 2022, 34 (19), e2200905.
- In the Discussion, the difference between PG Carbonisata and PGC-CDs needs to be elaborated.
Response: We thank the reviewer for reading our manuscript and providing us with useful comments, and our response is as follows: In traditional Chinese medicine, PGC is a kind of traditional medicine prepared by high temperature processing. Compared with PG, PGC contains fewer volatile components, and its pharmacological activity has changed accordingly. However, the material basis of its efficacy is not clear at present. Some scholars have found that the compounds contained in herbal medicines in the process of high-temperature processing can be transformed into carbon dots through dehydration, calcination, and carbonization, which have different biological activities from the original medicinal materials [1-3]. Therefore, in this study, we successfully obtained PGC-CDs from PGC, and discovered PGC-CDs was approximately spherical, with an average particle size of 2.3 nm, and mainly contained C, O, N elements, as well as carboxyl, hydroxyl, amino and other functional groups, indicated that PGC-CDs have better water solubility, more uniform particle size and new pharmacological activity different from the original medicinal compound. (Please see the revised manuscript, line 317-328).
References
- Luo, W. K.; Zhang, L. L.; Yang, Z. Y.; Guo, X. H.; Wu, Y.; Zhang, W.; Luo, J. K.; Tang, T.; Wang, Y., Herbal medicine derived carbon dots: synthesis and applications in therapeutics, bioimaging and sensing. Journal of nanobiotechnology 2021, 19 (1), 320.
- Gudimella, K. K.; Gedda, G.; Kumar, P. S.; Babu, B. K.; Yamajala, B.; Rao, B. V.; Singh, P. P.; Kumar, D.; Sharma, A., Novel synthesis of fluorescent carbon dots from bio-based Carica Papaya Leaves: Optical and structural properties with antioxidant and anti-inflammatory activities. Environmental research 2022, 204 (Pt A), 111854.
- Zhao, W.-B.; Wang, R.-T.; Liu, K.-K.; Du, M.-R.; Wang, Y.; Wang, Y.-Q.; Zhou, R.; Liang, Y.-C.; Ma, R.-N.; Sui, L.-Z.; Lou, Q.; Hou, L.; Shan, C.-X., Near-infrared carbon nanodots for effective identification and inactivation of Gram-positive bacteria. Nano Research 2022, 15 (3), 1699-1708.
- Some recent published papers about herbal medicine-CDs are suggested to be referred in this manuscript: Nano Research. 15, 9274–9285 (2022). Nano Research. 15, 1699–1708 (2022). Journal of Nanobiotechnology 19, 320 (2021).
Response: Thank you for providing high-level research papers on herbal CDs. We have carefully studied these studies and obtained the latest research progress in the field of herbal CDs, which will be conducive to our further research. At the same time, we found that these studies have a significant guiding role for this study, so we referred to these documents in the manuscript (Please see the revised manuscript, line 58-59, 323).
References
- Luo, W.; Zhang, L.; Li, X.; Zheng, J.; Chen, Q.; Yang, Z.; Cheng, M.; Chen, Y.; Wu, Y.; Zhang, .; Tang, T.; Wang, Y., Green functional carbon dots derived from herbal medicine ameliorate blood—brain barrier permeability following traumatic brain injury. Nano Research 2022, 15 (10), 9274-9285.
- Zhao, W.-B.; Wang, R.-T.; Liu, K.-K.; Du, M.-R.; Wang, Y.; Wang, Y.-Q.; Zhou, R.; Liang, Y.-C.; Ma, R.-N.; Sui, L.-Z.; Lou, Q.; Hou, L.; Shan, C.-X., Near-infrared carbon nanodots for effective identification and inactivation of Gram-positive bacteria. Nano Research 2022, 15 (3), 1699-1708.
- Luo, W. K.; Zhang, L. L.; Yang, Z. Y.; Guo, X. H.; Wu, Y.; Zhang, W.; Luo, J. K.; Tang, T.; Wang, Y., Herbal medicine derived carbon dots: synthesis and applications in therapeutics, bioimaging and sensing. Journal of nanobiotechnology 2021, 19 (1), 320.
- The manuscript has a considerable amount of unprofessional boasting that needs to be toned.
Response: Thank for your kind reminding. We have adjusted the content of the manuscript to ensure its preciseness. The specific content is described below: this study tries to break through the gap in this field (Please see the revised manuscript, line 86-87). This study provided a reference for the establishment of liver damage model induced by hyperbilirubinemia. More importantly, this study revealed that the control of inflammation and oxidative stress may be a potential strategy to reduce the harm of hyperbilirubinemia. However, this study is only a preliminary exploration of the effects and mechanisms of PGC-CDs in the treatment of hyperbilirubinemia and its induced liver damage, and further studies are needed to clarify the deeper potential mechanisms of these effects (line 388-394).
Considering the reviewer’s suggestion, we tried our best to improve the manuscript and made changes in the revised manuscript. Special thanks to you for your good comments.

Reviewer 2 Report
The authors of the manuscript have presented a novel biologically active compound derived from the plant Platycodon grandiflorum and its potential to treat hyperbilirubinemia and liver damage.
I have tested the manuscript using Plagscan software to evaluate the similarity of the proposed research to previously published ones. The obtained score was 11.9% indicating the novelty of the manuscript.
The work is new, and well organized, and the presentation of results is acceptable, still, some improvements should be done.
1. The authors have used several analytical techniques for the structural characterization of a new substance from plant extract. Still, more data are missing. Nuclear magnetic resonance (NMR) and mass spectrometry (MS) are the analytical tools that are routinely used to obtain structural data sets due to their versatility, accessibility, and unique strengths. More instrumental techniques use more reliable information is obtained. The authors haven't used those techniques to characterize the new compound. Those data would give more useful information on characterization than techniques such as UV/VIS. If it is possible, I highly recommend evaluating the novel compounds at least using MS/MS. Using this sophisticated technique molecular mass and fragmentation pattern of the compound can be obtained. Many laboratories offer MS, MS/MS or NMR scanning. These data would be a valuable contribution to your research.
2. Define abbreviations or acronyms on their first occurrence (lines: 96, 98, 118 …).
3. Define abbreviations in Figure captions (line 129).
4. The manuscript needs to be improved as I have found many typos (lines: 133, 134, 162, 163, 164, 165, 167, 175, 201, 230 ….). Please correct.
Author Response
Responses to the reviewer:
Comments and Suggestions for Authors
The authors of the manuscript have presented a novel biologically active compound derived from the plant Platycodon grandiflorum and its potential to treat hyperbilirubinemia and liver damage.
I have tested the manuscript using Plagscan software to evaluate the similarity of the proposed research to previously published ones. The obtained score was 11.9% indicating the novelty of the manuscript.
The work is new, and well organized, and the presentation of results is acceptable, still, some improvements should be done.
Response: We sincerely thank the reviewer for the valuable feedback that we have used to improve the quality of our manuscript. We have carefully adopted the suggestion of Reviewer and tried our best to improve and made some changes in the manuscript.
- The authors have used several analytical techniques for the structural characterization of a new substance from plant extract. Still, more data are missing. Nuclear magnetic resonance (NMR) and mass spectrometry (MS) are the analytical tools that are routinely used to obtain structural data sets due to their versatility, accessibility, and unique strengths. More instrumental techniques use more reliable information is obtained. The authors haven't used those techniques to characterize the new compound. Those data would give more useful information on characterization than techniques such as UV/VIS. If it is possible, I highly recommend evaluating the novel compounds at least using MS/MS. Using this sophisticated technique molecular mass and fragmentation pattern of the compound can be obtained. Many laboratories offer MS, MS/MS or NMR scanning. These data would be a valuable contribution to your research.
Response: We thank the reviewer for reading our manuscript and providing us with useful comments. The structure and components of CDs extracted from herbal medicine are very complex, and its structural formula cannot be obtained at present, so it is difficult to analyze them by nuclear magnetic resonance and mass spectrometry with small molecules. MS/MS analysis in related literature [1,2] of carbon dots were only used to ensure the functional groups, and it is difficult to analyse the structural composition of carbon dots through complex and fragmented ionic fragments. The lateral characterization of CDs using Fourier transform infrared, Ultraviolet–visible, X-ray photoelectron spectroscopy and other instruments is currently a common characterization method in the field of CDs [3,4].
References
- Xu, Y.; Wang, B.; Zhang, M.; Zhang, J.; Li, Y.; Jia, P.; Zhang, H.; Duan, L.; Li, Y.; Li, Y.; Qu, X.; Wang, S.; Liu, D.; Zhou, W.; Zhao, H.; Zhang, H.; Chen, L.; An, X.; Lu, S.; Zhang, S., Carbon Dots as a Potential Therapeutic Agent for the Treatment of Cancer-Related Anemia. Advanced materials (Deerfield Beach, Fla.) 2022, 34 (19), e2200905.
- Yao, L.; Zhao, M. M.; Luo, Q. W.; Zhang, Y. C.; Liu, T. T.; Yang, Z.; Liao, M.; Tu, P.; Zeng, K. W., Carbon Quantum Dots-Based Nanozyme from Coffee Induces Cancer Cell Ferroptosis to Activate Antitumor Immunity. ACS nano 2022, 16 (6), 9228-9239.
- Luo, W.; Zhang, L.; Li, X.; Zheng, J.; Chen, Q.; Yang, Z.; Cheng, M.; Chen, Y.; Wu, Y.; Zhang, .; Tang, T.; Wang, Y., Green functional carbon dots derived from herbal medicine ameliorate blood—brain barrier permeability following traumatic brain injury. Nano Research 2022, 15 (10), 9274-9285.
- Gudimella, K. K.; Gedda, G.; Kumar, P. S.; Babu, B. K.; Yamajala, B.; Rao, B. V.; Singh, P. P.; Kumar, D.; Sharma, A., Novel synthesis of fluorescent carbon dots from bio-based Carica Papaya Leaves: Optical and structural properties with antioxidant and anti-inflammatory activities. Environmental research 2022, 204 (Pt A), 111854.
- Define abbreviations or acronyms on their first occurrence (lines: 96, 98, 118 …).
Response: We sincerely appreciate the reviewer for careful reading, and apologize for the errors in the manuscript. The specific content is described below: Transmission electron microscope (TEM) (Please see the revised manuscript, line 97). X-ray Diffraction (XRD) (line 100). Ultraviolet–visible (UV-Vis) (line 105). X-ray photoelectron spectroscopy (XPS) (line 122). Fourier transform infrared (FTIR) (line 131). bilirubin group (Bil group) (line 179).…detecting levels of direct bilirubin (DBIL), indirect bilirubin (IBIL), total bilirubin (TBIL), total bile acid (TBA), alanine aminotransferase (ALT) and aspartate aminotransferase (AST) (line 211-213). The levels of interleukin-6 (IL-6) and tumor necrosis factor-α (TNF-α) (line 250). Superoxide dismutase (SOD) (line 273). malondialdehyde (MDA) (line 279). Glutathione (GSH) (line 284). Catalase (CAT) (line 291). Hematoxylin and eosin (H&E) staining (line 302).
- Define abbreviations in Figure captions (line 129).
Response: We sincerely thank the reviewer for careful reading. As suggested by the reviewer, we have corrected the errors. The specific content is described below: Ultraviolet–visible (UV-Vis) spectrum (Please see the revised manuscript, line 120). Fourier transform infrared (FTIR) spectrum (line 120). X-ray photoelectron spectroscopy (XPS) spectrum of PGC-CDs (line 133). Effects of PGC-CDs on inflammatory factors interleukin-6 (IL-6) (A) and tumor necrosis factor-α (TNF-α) (B) in serum of mice with hyperbilirubinemia model. Effects of PGC-CDs on superoxide dismutase (SOD) (C), malondialdehyde (MDA) (D), glutathione (GSH) (E) and catalase (CAT) (line 265-267).
- The manuscript needs to be improved as I have found many typos (lines: 133, 134, 162, 163, 164, 165, 167, 175, 201, 230 ….). Please correct.
Response: We were really sorry for our careless mistakes. In the resubmitted manuscript, the typos were revised. Thanks for your correction. The specific content is described below: ...and could reduce the oxidative damage of cells caused by H2O2 (Please see the revised manuscript, line 27). it is necessary to establish an overall plan (line 81). When PGC-CDs were not given, the cell viability was calculated to be 100% according to the cell viability formula, which was the normal viability of RAW264.7 cells. As a reference, when the value of cell viability was less than 100%, indicating that it could inhibit cell proliferation. when the value of cell viability was equal to or more than 100% (line 138-142). As shown in the Figure 3C, compared with the model group (26.58±3.45 U/mgprot), 250, 125 and 62.5 μg/mL of PGC-CDs could significantly increase the SOD activity (45.15±5.47 U/mgprot, 60.78±6.61 U/mgprot, 57.14±9.11 U/mgprot, P < 0.01). As shown in Figure 3D, compared with the control group (2±0.45 nmol/mgprot), the MDA content in cells of the model group (17.78±2.12 nmol/mgprot) increased significantly (P < 0.01). 250, 125 and 62.5 μg/mL of PGC-CDs (12.68±1.63, 8.98±2.60, 11.16±3.15 nmol/mgprot, P < 0.01) significantly inhibited the increase of MDA content in cells caused by H2O2 (line 166-173). After intraperitoneal injection of bilirubin (line 178). It was worth noting that Tarlov scores of PGC-CDs groups were significantly higher (line 203). Tarlov scores of mice (line 206). disordered hepatic cord (line 306). high-power microscope (line 308). apparently (line 310). inhibit vascular activity (line 368).
Considering the reviewer’s suggestion, we have improved the manuscript. Special thanks to you for your good comments.

Reviewer 3 Report
Manuscript ID: molecules-2151052
Title: A novel drug with potential to treat hyperbilirubinemia and prevent liver damage induced by hyperbilirubinemia: Carbon dots derived from Platycodon grandiflorum
The authors described the synthesis and characterization of carbon dots from Platycodon grandiflorum. The obtained nanoparticles were tested in the treatment of hyperbilirubinemia. In my opinion, the manuscript would be more suitable for a medical journal than for the nanochemistry section. From a chemistry approach, the manuscript must be improved starting from the definitions and fundamental notions.
I have the following observations:
1. “PG-based carbon dots” can not be “a substance” since a substance is defined as having “definite chemical constitution”. The chemical constitution is not definite and a chemical formula can not be assigned.
2. “discovered and extracted a new substance from PG” – the CDs were not “discovered and extracted”, they were obtained by thermal treatment. CDs don't exist in PG, they were further obtained from plants.
3. “PGC-CDs, a novel CDs was prepared by one-step pyrolysis” – pyrolysis is defined as “the heating of an organic material, such as biomass, in the absence of oxygen”. The thermal treatment was not done in the absence of oxygen, even if the crucible was covered. The presence of an inert gas has been necessary. Furthermore, “Biomass pyrolysis is usually conducted at or above 500 °C, providing enough heat to deconstruct the strong bio-polymers mentioned above.”. The temperature used in this study seems to be too low.
4. Which is the difference between PGC-CDs and other CDs obtained from plants? This is an important question.
5. Were the CDs obtained as a solid material or as a suspension, in water? It is also an important issue and must be stated. The manner in which the authors refer to the obtained product should be appropriate to the form in which it exists
6. “CDs, as a novel type of carbon nanotube” – CDs and CNTs are different issues (see, for example, “Carbon Dots: A New Type of Carbon-Based Nanomaterial with Wide Applications”, https://pubs.acs.org/doi/pdf/10.1021/acscentsci.0c01306)
7. “safety and antioxidation of PGC-CDs was evaluated in cells” – antioxidation or antioxidation activity?
8. The CDs synthesis is not described clearly enough to be reproducible.
9. Were the spectra obtained for powders or solutions? This information should be provided more clearly in the manuscript.
10. References are required for each of the methods used.
11. References for FTIR spectrum assignments are necessary.
12. “The composition and coordination of PGC-CDs were obtained by observing XPS” – How do you mean by “coordination”?
13. In “3. Discussion” the authors must discuses their experimental results, not the information from literature. So, the first paragraph can be removed. The same observation for other theoretical considerations in this section (which are too numerous reported to the original results).
14. The typos must be corrected and English must be revised.
Author Response
Responses to the reviewer:
Title: A novel drug with potential to treat hyperbilirubinemia and prevent liver damage induced by hyperbilirubinemia: Carbon dots derived from Platycodon grandiflorum
The authors described the synthesis and characterization of carbon dots from Platycodon grandiflorum. The obtained nanoparticles were tested in the treatment of hyperbilirubinemia. In my opinion, the manuscript would be more suitable for a medical journal than for the nanochemistry section. From a chemistry approach, the manuscript must be improved starting from the definitions and fundamental notions.
Response: Thank you for your constructive suggestions. Carbon dots, as a carbon-based nanomaterial, are a complex mixture of small molecules, polymers and CDs. Their structure is very complex, and the structural formula cannot be given. However, there are unique characterization methods in the field of carbon dots, such as FTIR, UV-Vis, XPS, etc., which are recognized as the characterization methods of carbon dots [1-3]. Molecules magazine contains a host of excellent papers on the synthesis and applications of functional carbon quantum dots, so this manuscript is just right for Molecules.
References
- Xu, Y.; Wang, B.; Zhang, M.; Zhang, J.; Li, Y.; Jia, P.; Zhang, H.; Duan, L.; Li, Y.; Li, Y.; Qu, X.; Wang, S.; Liu, D.; Zhou, W.; Zhao, H.; Zhang, H.; Chen, L.; An, X.; Lu, S.; Zhang, S., Carbon Dots as a Potential Therapeutic Agent for the Treatment of Cancer-Related Anemia. Advanced materials (Deerfield Beach, Fla.) 2022, 34 (19), e2200905.
- Luo, W.; Zhang, L.; Li, X.; Zheng, J.; Chen, Q.; Yang, Z.; Cheng, M.; Chen, Y.; Wu, Y.; Zhang, .; Tang, T.; Wang, Y., Green functional carbon dots derived from herbal medicine ameliorate blood—brain barrier permeability following traumatic brain injury. Nano Research 2022, 15 (10), 9274-9285.
- Gudimella, K. K.; Gedda, G.; Kumar, P. S.; Babu, B. K.; Yamajala, B.; Rao, B. V.; Singh, P. P.; Kumar, D.; Sharma, A., Novel synthesis of fluorescent carbon dots from bio-based Carica Papaya Leaves: Optical and structural properties with antioxidant and anti-inflammatory activities. Environmental research 2022, 204 (Pt A), 111854.
I have the following observations:
- “PG-based carbon dots” can not be “a substance” since a substance is defined as having “definite chemical constitution”. The chemical constitution is not definite and a chemical formula can not be assigned.
Response: Thank you for your careful and kind reminding. CDs is a general term for zero-dimensional carbon nanoparticles, which is composed of ultra-fine, dispersed carbon nanoparticles with a size less than 10 nm. Generally, surface atoms of CDs are connected with a large number of oxygen-containing functional groups, so the structure and composition of CDs are very complex. We cannot give chemical formula and define them as substances. We apologize for the mistakes in expression. We changed the expression in the manuscript. The specific content is described below: In this study, a novel carbon dots (CDs), PG-based CDs (PGC-CDs) (Please see the revised manuscript, line 19).
- discovered and extracted a new substance from PG” – the CDs were not “discovered and extracted”, they were obtained by thermal treatment. CDs don't exist in PG, they were further obtained from plants.
Response: We are grateful for the reviewer for bringing this to our attention. We have changed the expression in the revised manuscript. The specific content is described below: we first obtained PGC-CDs from carbonized PG (Please see the revised manuscript, line 88).
- “PGC-CDs, a novel CDs was prepared by one-step pyrolysis” – pyrolysis is defined as “the heating of an organic material, such as biomass, in the absence of oxygen”. The thermal treatment was not done in the absence of oxygen, even if the crucible was covered. The presence of an inert gas has been necessary. Furthermore, “Biomass pyrolysis is usually conducted at or above 500 °C, providing enough heat to deconstruct the strong bio-polymers mentioned above.”. The temperature used in this study seems to be too low.
Response: Thank you for your kind guidance. "Bottom-up" is one of the preparation methods of CDs. This method mainly uses small molecular organic matter or polymer as carbon source, and forms CDs through dehydration, condensation, carbonization and other processes. 200℃ to 400℃ is the common calcination temperature for preparing CDs. At this temperature, the reaction conditions are relatively mild, Generally, the CDs obtained have no obvious crystal structure [1]. The carbon core structure may be either amorphous carbon or nanoparticle formed by non-conjugated polymer crosslinking [2], which can retain the active groups. However, if the pyrolysis temperature is further increased to 500℃, large and heterogeneous carbon nanoparticles with poor solubility in water can be obtained, with a particle size of several hundred nanometers, which may come from aggregation and cause the loss of active groups. Thus, the temperature of the pyrolysis was really too high. In this study, the calcination temperature is 350 ℃, which could not be described by pyrolysis. Therefore, the description should be revised. The specific content is described below: prepared from PG via high-temperature calcination (Please see the revised manuscript, line 20). PGC-CDs, a novel CDs was prepared by high-temperature calcination (line 519).
References
- Zhu, C.; Zhai, J.; Dong, S., Bifunctional fluorescent carbon nanodots: green synthesis via soy milk and application as metal-free electrocatalysts for oxygen reduction. Chemical communications (Cambridge, England) 2012, 48 (75), 9367-9.
- Zhu, S.; Song, Y.; Shao, J.; Zhao, X.; Yang, B., Non-Conjugated Polymer Dots with Crosslink-Enhanced Emission in the Absence of Fluorophore Units. Angewandte Chemie (International ed. in English) 2015, 54 (49), 14626-37.
- Which is the difference between PGC-CDs and other CDs obtained from plants? This is an important question.
Response: It is a valid point that the difference between PGC-CDs and other CDs was observed in their structural characterization, optical features and different biological activities [1-4], which may obtain to their precursors consisted with different compounds and in-depth reasons are to be explored further in subsequent studies. Moreover, the difference is closely related to the precursor [5]. The carbon skeletons condensed after heating are different from the surface groups.
The main elements of plant part-derived CDs are C, O, H, and N atoms, which present in various functional groups and provide good water solubility. When these carbon sources are used for the synthesis of CDs, most of them do not need the addition of a substance for doping, but rather self-dope [6]. Different sources of CDs self-doped elements may vary. For example, there is sp hybridization carbons in Crinis Carbonisatus CDs (CrCi-CDs), which possess abundant nitrogen-doped [7]. Phosphorus and nitrogen co-doped CDs were obtained by using almond as carbon source [8]。Zhang et al. prepared N- and S-CDs using scallion leaves [9].
Most plant part-derived CDs exhibit absorbance in the range of 280–360 nm, but there are certain differences in the position of absorption peaks. The CDs derived from date kernel [10], groundnuts [11], and pineapple peel [12] exhibited absorbance peaks at 275, 279, and 280 nm, extending until the near-visible range associated with the aromatic C=C bonds of the π-π* transition. The UV-vis absorbance of CDs from lemon juice showed the absorption peak at 280 nm with a tail expanding to the visible region associated with the conjugated C=O bond transition [13].
CDs prepared by different methods, reaction conditions and raw materials often have different properties, CrCi-CDs performs neuroprotective effect on cerebral ischemia and reperfusion injury [7]. Specific selection and high sensitivity of almond-derived carbon dots to Fe3+ in aqueous solution [8]. CDs synthesized from soybean milk has not only good photoluminescence properties, but also good electrocatalytic activity for oxygen reduction reaction [14]. Wang et al. prepared ethanol-soluble CDs (E-CDs) and water-soluble CDs (W-CDs) with papaya flesh powder [15].
References
- Luo, W. K.; Zhang, L. L.; Yang, Z. Y.; Guo, X. H.; Wu, Y.; Zhang, W.; Luo, J. K.; Tang, T.; Wang, Y., Herbal medicine derived carbon dots: synthesis and applications in therapeutics, bioimaging and sensing. Journal of nanobiotechnology 2021, 19 (1), 320.
- Li, D.; Xu, K. Y.; Zhao, W. P.; Liu, M. F.; Feng, R.; Li, D. Q.; Bai, J.; Du, W. L., Chinese Medicinal Herb-Derived Carbon Dots for Common Diseases: Efficacies and Potential Mechanisms. Frontiers in pharmacology 2022, 13, 815479.
- Luo, W.; Zhang, L.; Li, X.; Zheng, J.; Chen, Q.; Yang, Z.; Cheng, M.; Chen, Y.; Wu, Y.; Zhang, .; Tang, T.; Wang, Y., Green functional carbon dots derived from herbal medicine ameliorate blood—brain barrier permeability following traumatic brain injury. Nano Research 2022, 15 (10), 9274-9285.
- Zhao, W.-B.; Wang, R.-T.; Liu, K.-K.; Du, M.-R.; Wang, Y.; Wang, Y.-Q.; Zhou, R.; Liang, Y.-C.; Ma, R.-N.; Sui, L.-Z.; Lou, Q.; Hou, L.; Shan, C.-X., Near-infrared carbon nanodots for effective identification and inactivation of Gram-positive bacteria. Nano Research 2022, 15 (3), 1699-1708.
- Das, R.; Bandyopadhyay, R.; Pramanik, P., Carbon quantum dots from natural resource: A review. Materials Today Chemistry 2018, 8, 96-109.
- Meng W., Bai X., Wang B., Liu Z., Lu S., Yang B. Biomass-derived carbon dots and their applications. Energy Environ. Mater 2019, 2, 172–192.
- Zhang, Y.; Wang, S.; Lu, F.; Zhang, M.; Kong, H.; Cheng, J.; Luo, J.; Zhao, Y.; Qu, H., The neuroprotective effect of pretreatment with carbon dots from Crinis Carbonisatus (carbonized human hair) against cerebral ischemia reperfusion injury. Journal of nanobiotechnology 2021, 19 (1), 257.
- Dai, L.T., Synthesis, characterization and application of fluorescent carbon points derived from natural substances.Taiyuan:Shanxi University 2018 (in Chinese).
- Zhang Z., Hu B., Zhuang Q., Wang Y., Luo X., Xie Y., Zhou D. Green synthesis of fluorescent nitrogen–sulfur Co-doped carbon dots from scallion leaves for hemin sensing. Analytical letters 2020, 53, 1704–1718.
- Amin N., Afkhami A., Hosseinzadeh L., Madrakian T. Green and cost-effective synthesis of carbon dots from date kernel and their application as a novel switchable fluorescence probe for sensitive assay of Zoledronic acid drug in human serum and cellular imaging. Anal. Chim. Acta 2018, 1030, 183–193.
- Roshni V., Misra S., Santra M.K., Ottoor D. One pot green synthesis of C-dots from groundnuts and its application as Cr (VI) sensor and in vitro bioimaging agent. J. Photochem. Photobiol. A Chem 2019, 373, 28–36.
- Vandarkuzhali S.A.A., Natarajan S., Jeyabalan S., Sivaraman G., Singaravadivel S., Muthusubramanian S., Viswanathan B. Pineapple peel-derived carbon dots: Applications as sensor, molecular keypad lock, and memory device. ACS Omega 2018, 3, 12584–12592.
- Hoan B.T., Thanh T.T., Tam P.D., Trung N.N., Cho S., Pham H. A green luminescence of lemon derived carbon quantum dots and their applications for sensing of V5+ ions. Mater. Sci. Eng. B 2019, 251:114455.
- Fan, Z.; Li, S.; Yuan, F., Flourescent graphene quantum dots forbiosensing and bioimaging. RSC Advances 2015, 5 (25), 19773-19789.
- Wang N., Wang Y., Guo T., Yang T., Chen M.-L., Wang J.-H. Green preparation of carbon dots with papaya as carbon source for effective fluorescent sensing of Iron (III) and Escherichia Coli. Biosens. Bioelectron 2016, 85, 68–75.
- Were the CDs obtained as a solid material or as a suspension, in water? It is also an important issue and must be stated. The manner in which the authors refer to the obtained product should be appropriate to the form in which it exists
Response: Thanks for the reviewer’s reasonable doubt. PGC-CDs were not obtained in water in the form of solid material or suspension. PGC-CDs were CDs with particle size ranging from 1.2 nm to 3.6nm, so they existed in water in the form of colloidal solution of 1-100 nm, with the Tyndall effect of the colloidal solution at the nanoscale. Although it was not true solution, according to our observation, PGC-CDs solution was not easy to aggregate and sediment, and had good solubility and stability, which was related to the hydrophilic functional groups.
- “CDs, as a novel type of carbon nanotube” – CDs and CNTs are different issues (see, for example, “Carbon Dots: A New Type of Carbon-Based Nanomaterial with Wide Applications”, https://pubs.acs.org/doi/pdf/10.1021/acscentsci.0c01306)
Response: We sincerely thank the reviewer for careful reading and patient guidance, we were really sorry for our careless mistakes. Carbon dots are considered to be zero-dimensional materials, graphene is considered to be two-dimensional materials, carbon nanotubes are considered to be three-dimensional materials, usually derived from graphene. “CDs, as a novel type of carbon nanotube” was a description error. According to your suggestion, we have deleted the first paragraph of the discussion which contained the wrong description in the revised manuscript.
- “safety and antioxidation of PGC-CDs was evaluated in cells” – antioxidation or antioxidation activity?
Response: Thank you for your detailed questions. It should be antioxidant activity, so the original description was revised as the safety and antioxidant activity of PGC-CDs was evaluated by RAW264.7 cells (Please see the revised manuscript, line 23). the safety and antioxidant activity of PGC-CDs were evaluated in cells (line 89).
- The CDs synthesis is not described clearly enough to be reproducible.
Response: We thank you for the critical comments and helpful suggestion. We have changed the expression in the revised manuscript. The differences between batches of PGC-CDs prepared by this process were small and easy to reproduce. The specific content is described below: The preparation process of PGC-CDs can be divided into carbonization, boiling and purification (Figure 8). PG was weighed and placed in a clean and dry crucible, then it was sealed with aluminum foil and put in a muffle furnace for calcination. The temperature process of muffle furnace was as follows: the calcined temperature was increased to 70℃ within 5 min, after maintaining at 70℃ for 30 min, the temperature was increased from 70℃ to 350℃ within 25 min, and maintained at 350℃ for 1 h. After the temperature of muffle furnace dropped to 40℃, the PGC was further taken and crushed. Then the PGC power was added to thirtyfold DW and boiled at 100ºC twice for 1 h, during which glass rod was used to stir evenly. The decoction was combined and filtered with 0.22 μm microporous membrane, and the filtrate was collected and concentrated to obtain 1 g/ mL PGC solution. The concentrated solution was transferred into a dialysis membrane with a molecular weight cut-off of 1000 Da, the dialysis membrane was placed in a beaker with DW for dialyzing for 7 days, during which the DW was replaced every 4 hours. Until the liquid outside the dialysis membrane was transparent, the solution was removed from the dialysis membrane and placed in a 4ºC refrigerator for future use (Please see the revised manuscript, line 419-434).
- Were the spectra obtained for powders or solutions? This information should be provided more clearly in the manuscript.
Response: Thank you for pointing this out. This study used PGC-CDs solution to obtain the ultraviolet, fluorescence and infrared spectra of PGC-CDs. This explanation is supplemented in the manuscript. The specific content is described below:
The optical characteristics of PGC-CDs solution was analyzed by fluorescence spectroscopy and UV-Vis spectrophotometer. FTIR spectroscopy was used to analyze the surface group information in PGC-CDs solution (Please see the revised manuscript, line 440-442).
- References are required for each of the methods used.
Response: As suggested by the reviewer, we have added more references to support this idea. The specific content and references are described below: TEM and HR-TEM were used to observe the morphology, particle size distribution and lattice spacing of PGC-CDs [1] (Please see the revised manuscript, line 439). CCK-8 assay was used to evaluate the cytotoxicity of PGC-CDs on RAW264.7 cells [2] (line 447). The oxidative stress model of RAW264.7 cells induced by H2O2 was used to evaluate the antioxidant activity of PGC-CDs in cells [3] (line 462).
References
- Wang, L.; Wang, Y.; Xu, T.; Liao, H.; Yao, C.; Liu, Y.; Li, Z.; Chen, Z.; Pan, D.; Sun, L.; Wu, M., Gram-scale synthesis of single-crystalline graphene quantum dots with superior optical properties. Nature communications 2014, 5, 5357.
- Zhao, G.; Hu, C.; Xue, Y., In vitro evaluation of chitosan-coated liposome containing both coenzyme Q10 and alpha-lipoic acid: Cytotoxicity, antioxidant activity, and antimicrobial activity. Journal of cosmetic dermatology 2018, 17 (2), 258-262.
- Li, J.; Li, Y.; Li, Y.; Yang, Z.; Jin, H., Physicochemical Properties of Collagen from Acaudina Molpadioides and Its Protective Effects against H(2)O(2)-Induced Injury in RAW264.7 Cells. Marine drugs 2020, 18 (7).
- References for FTIR spectrum assignments are necessary.
Response: We sincerely appreciate the valuable suggestion. We have checked the literature carefully and added more references on FTIR spectrum assignments into the Results part in the revised manuscript (Please see the revised manuscript, line 113, 115). The specific content and references are described below:
The characteristic peak at 3440 cm-1 suggested the possibility of stretching vibration of -O-H bond or -N-H bond, while the absorption peak at 2920 cm-1 and 2851 cm-1 were caused by the stretching vibration of C-H bond of CH2 on PGC-CDs surface, and the characteristic peak at 1629 cm-1 was generally considered to be caused by C=O bond[1], the stretching vibration peak of C-N was usually located at the characteristic peak of 1384 cm-1, and the characteristic peak of 1057 cm-1 was attributed to the stretching vibration of C-O-C bond[2-4].
References
- Zhang, M.; Cheng, J.; Zhang, Y.; Kong, H.; Wang, S.; Luo, J.; Qu, H.; Zhao, Y., Green synthesis of Zingiberis rhizoma-based carbon dots attenuates chemical and thermal stimulus pain in mice. Nanomedicine (London, England) 2020, 15 (9), 851-869.
- Zhang, J. H.; Niu, A.; Li, J.; Fu, J. W.; Xu, Q.; Pei, D. S., In vivo characterization of hair and skin derived carbon quantum dots with high quantum yield as long-term bioprobes in zebrafish. Scientific reports 2016, 6, 37860.
- Mewada, A.; Pandey, S.; Shinde, S.; Mishra, N.; Oza, G.; Thakur, M.; Sharon, M.; Sharon, M., Green synthesis of biocompatible carbon dots using aqueous extract of Trapa bispinosa peel. Materials science & engineering. C, Materials for biological applications 2013, 33 (5), 2914-7.
- Zhao, Y.; Zhang, Y.; Kong, H.; Zhang, M.; Cheng, J.; Wu, J.; Qu, H.; Zhao, Y., Carbon Dots from Paeoniae Radix Alba Carbonisata: Hepatoprotective Effect. International journal of nanomedicine 2020, 15, 9049-9059.
- “The composition and coordination of PGC-CDs were obtained by observing XPS” – How do you mean by “coordination”?
Response: Thanks for your careful checks. We are sorry for our carelessness. Based on your comments, we have made the correction to make the word harmonized within the whole manuscript. The specific content is described below: The element composition and bonding actions of PGC-CDs were obtained by observing XPS (Please see the revised manuscript, line 121). XPS was used to analyze the element composition and bonding actions of PGC-CDs (line 442-443).
- In “3. Discussion” the authors must discuss their experimental results, not the information from literature. So, the first paragraph can be removed. The same observation for other theoretical considerations in this section (which are too numerous reported to the original results).
Response: Thank you for your thoughtful suggestions. We have re-written this part according to the reviewer’s suggestion (Please see the revised manuscript, line 317-394). The specific content is described below:
In traditional Chinese medicine, PGC is a kind of traditional medicine prepared by high temperature processing. Compared with PG, PGC contains fewer volatile components, and its pharmacological activity has changed accordingly. However, the material basis of its efficacy is not clear at present. Some scholars have found that the compounds contained in herbal medicines in the process of high-temperature processing can be transformed into carbon dots through dehydration, calcination, and carbonization, which have different biological activities from the original medicinal materials [28-30]. Therefore, in this study, we successfully obtained PGC-CDs from PGC, and discovered PGC-CDs was approximately spherical, with an average particle size of 2.3 nm, and mainly contained C, O, N elements, as well as carboxyl, hydroxyl, amino and other functional groups, indicated that PGC-CDs have better water solubility, more uniform particle size and new pharmacological activity different from the original medicinal compound. In addition, its safety was verified by cytotoxicity assay. Meanwhile, the antioxidant activity of PGC-CDs in cells was confirmed.
The etiology of hyperbilirubinemia is very complex, and it is one of the most common disease of newborns, and also occurs in people such as hepatitis, cirrhosis, sepsis, liver transplantation and heart surgery [31]. Studies [31-37] have shown that hyperbilirubinemia and liver injury interact with each other. When liver clearance is low [38], hyperbilirubinemia may occur. Meanwhile, patients with hyperbilirubinemia may need liver transplantation [39, 40]. Therefore, liver protection is particularly important in the treatment of hyperbilirubinemia. It was found that PGC-CDs had a good inhibitory effect on hyperbilirubinemia by studying the body weight, food intake, survival rate and neurological function of mice. Moreover, some therapeutic effects of medium and low doses of PGC-CDs were better than that of high doses of PGC-CDs, which may be due to the particle density of PGC-CDs in high dose was large, the aggregation and sedimentation were increased, the dispersion and fluidity were poor, and the activity was reduced [41].
The treatment of hyperbilirubinemia should solve the state of high levels of bilirubin as soon as possible, and reduce the content of bilirubin, which is toxic to humans. In this study, the serum biochemical results of DBIL, IBIL and TBIL increased sharply, indicating the successful establishment of hyperbilirubinemia model. PGC-CDs can significantly reduce the level of serum bilirubin, demonstrating that PGC-CDs can treat hyperbilirubinemia from the perspective of reducing the level of bilirubin so as to re-duce the damage of high levels of bilirubin to the body and mortality. Patients with hyperbilirubinemia often have higher levels of ALT and AST, and it is always necessary to require hepatoprotective therapy to prevent irreversible liver damage and avoid uncontrollable hyperbilirubinemia. ALT and AST increased sharply in the Bil group, indicating that hyperbilirubinemia can cause liver damage. It was a remarkable fact that all doses of PGC-CDs reduced the levels of TBA, ALT, and AST, suggesting that PGC-CDs had a protective effect on liver damage induced by hyperbilirubinemia. This result was particularly important because the protective effect of PGC-CDs on the liver was achieved without affecting levels of total bilirubin in plasma. Of course, pathological observations further confirmed this result.
Some chronic inflammatory can cause hyperbilirubinemia [42, 43]. The use of anti-inflammatory drugs can effectively reduce neonatal hyperbilirubinemia mortality and avoid nerve damage [17]. Levels of inflammatory factors in the Bil group increased sharply, suggesting that mice were in a state of severe inflammation, which inevitably affected liver function and bilirubin metabolism. It was noteworthy that all doses of PGC-CDs can significantly reduce levels of IL-6 and TNF-α, showing great anti-inflammatory activity. Inflammation caused by elevated bilirubin increases mortality from the disease. The results of this study suggested that PGC-CDs can inhibit levels of IL-6 and TNF-α in the treatment of hyperbilirubinemia and its associated liver damage, thereby reducing the mortality of hyperbilirubinemia. From the mechanism analysis, PGC-CDs can inhibit inflammation, increase hepatic blood flow, reduce edema, inhibit vascular activity, and stabilize cell membrane and lysosomal membrane. And it can also reduce tissue edema, reduce the damage of tissue cell structure, inhibit the release of inflammatory factors, so as to enhance the body's tolerance to hyperbilirubinemia.
In fact, oxidative stress reactions are common in the presence of excessive bilirubin [44, 45]. Levels of SOD, GSH, CAT and other substances involved in scavenging reactive oxygen free radicals in the body decrease, leading to the increase of the content of active oxygen free radicals in the body, which in turn cause oxidative stress and dam-age. The Level of MDA usually increases when liver cells or tissues are damaged. The results of this study showed that activity of SOD, the level of GSH and CAT of mice in the Bil group decreased significantly, while the level of MDA increased, suggesting that liver suffered more serious oxidative stress after hyperbilirubinemia. After PGC-CDs pre-treatment, SOD activity, the level of GSH and CAT increased obviously, while the level of MDA decreased significantly, indicating that PGC-CDs can effectively improve the antioxidant capacity of the body. From the mechanism analysis, PGC-CDs can antagonize the excitotoxic response of glutamate to hepatocytes, and can restore the reduced antioxidant enzymes in red blood cells to the normal level and play a protective role. At the same time, it can reduce the generation of reactive oxy-gen species and reduce the function of mitochondrial overload, so as to exert antioxidant effect.
Hyperbilirubinemia model induced by direct injection of bilirubin is a stable, economical and simple model [34]. This study provided a reference for the establishment of liver damage model induced by hyperbilirubinemia. More importantly, this study revealed that the control of inflammation and oxidative stress may be a potential strategy to reduce the harm of hyperbilirubinemia. However, this study is only a preliminary exploration of the effects and mechanisms of PGC-CDs in the treatment of hyperbilirubinemia and its induced liver damage, and further studies are needed to clarify the deeper potential mechanisms of these effects.
- The typos must be corrected and English must be revised.
Response: We were really sorry for our careless mistakes. In the resubmitted manuscript, the typos and English were revised. Thanks for your correction. The specific content is described below: ...and could reduce the oxidative damage of cells caused by H2O2 (Please see the revised manuscript, line 27). it is necessary to establish an overall plan (line 81). When PGC-CDs were not given, the cell viability was calculated to be 100% according to the cell viability formula, which was the normal viability of RAW264.7 cells. As a reference, when the value of cell viability was less than 100%, indicating that it could inhibit cell proliferation. when the value of cell viability was equal to or more than 100% (line 138-142). As shown in the Figure 3C, compared with the model group (26.58±3.45 U/mgprot), 250, 125 and 62.5 μg/mL of PGC-CDs could significantly increase the SOD activity (45.15±5.47 U/mgprot, 60.78±6.61 U/mgprot, 57.14±9.11 U/mgprot, P < 0.01). As shown in Figure 3D, compared with the control group (2±0.45 nmol/mgprot), the MDA content in cells of the model group (17.78±2.12 nmol/mgprot) increased significantly (P < 0.01). 250, 125 and 62.5 μg/mL of PGC-CDs (12.68±1.63, 8.98±2.60, 11.16±3.15 nmol/mgprot, P < 0.01) significantly inhibited the increase of MDA content in cells caused by H2O2 (line 166-173). After intraperitoneal injection of bilirubin (line 178). It was worth noting that Tarlov scores of PGC-CDs groups were significantly higher (line 203). Tarlov scores of mice (line 206). The increase of AST was significantly inhibited in the high and medium dose PGC-CDs groups (line 236-237). disordered hepatic cord (line 306). high-power microscope (line 308). apparently (line 310). inhibit vascular activity (line 368).
Considering the reviewer’s suggestion, we have studied comments carefully and have improved the manuscript. Special thanks to you for your good comments.

Round 2
Reviewer 1 Report
Thank you to the author's for their responses. That said, there are a few points that should be addressed:
1. The CDs prepared from other herbal medicines could not be used as reference for dose selection in this study. Authors need to provide reasonable experimental evidence for the choice of three doses.
2. The references about “Tarlov scores” in the hyperbilirubinemia must be added in the method section.
3. The results of HPLC of PGC-CDs must be added to illustrate that no small molecular components in the PGC-CDs solution.
